



# The MAPM (Mapping Air Pollution eMissions) method for inferring particulate matter emissions maps at city-scale from in situ concentration measurements: description and demonstration of capability

Brian Nathan[1,2], Stefanie Kremser[2], Sara Mikaloff-Fletcher[1], Greg Bodeker[2], Leroy Bird[2], Ethan Dale[2], Dongqi Lin[3], Gustavo Olivares[4], and Elizabeth Somervell[4]

[1]NIWA, Wellington, New Zealand
[2]Bodeker Scientific, Alexandra, New Zealand
[3]University of Canterbury, Christchurch, New Zealand
[4]NIWA, Auckland, New Zealand

**Correspondence:** Brian Nathan (Dr.Brian.Nathan@gmail.com)

**Abstract.** MAPM (Mapping Air Pollution eMissions) is a two-year project whose goal is to develop a method to infer particulate matter (PM) emissions maps from in situ PM concentration measurements. Central to the functionality of MAPM is an inverse model. The input of the inverse model includes a spatially-distributed prior emissions estimate and PM measurement time series from instruments distributed across the desired domain. Here we describe the construction of this inverse model, the mathematics underlying the retrieval of the resultant posterior PM emissions maps, the way in which uncertainties are traced through the MAPM processing chain, and plans for future development of the processing chain. To demonstrate the capability of the inverse model developed for MAPM, we use the $PM_{2.5}$ measurements obtained during a dedicated winter field campaign in Christchurch, New Zealand, in 2019 to infer $PM_{2.5}$ emissions maps on city scale. The results indicate a systematic overestimation in the prior emissions for Christchurch of at least 40–60%, which is consistent with some of the underlying assumptions used in the composition of the bottom-up emissions map used as the prior, highlighting the uncertainties in bottom-up approaches for estimating $PM_{2.5}$ emissions maps.

The paper also presents the results of two sets of observing system simulation experiments (OSSEs) that explore how measurement uncertainties affect the computation of the derived emissions maps, and the extent to which using emissions maps from one day as the prior for the next day improves the ability of the inversion system to characterize the emissions sources. We find in the first case that a smaller number of high-accuracy instruments performs significantly better than a higher number of low-accuracy instruments. In the second case, the results are ultimately inconclusive, showing the need for further investigations that are beyond the scope of this study.



# 1 Introduction

The growth of mega-cities from global urbanization has degraded urban air quality sufficiently to impede economic growth
and create a public health hazard (Adams et al., 2015). Emissions of particulate matter (PM), photochemically reactive gases,
and long-lived greenhouse gases, contribute to the urban environmental footprint with concomitant economic and social costs.
Recent research has demonstrated a link between air pollution levels and elevated susceptibility of the public to pulmonary
diseases (Anderson et al., 2012; Crinnion, 2017). Multiple studies have also characterized increases in hospitalizations for
cardiac and respiratory diseases that are directly correlated with increases in $PM_{2.5}$ concentrations (e.g. Dominici et al., 2006;
Zanobetti et al., 2009). With regard to the recent COVID-19 pandemic, Zhu et al. (2020) showed that a $10\ \mu g\ m^{-3}$ increase
in $PM_{2.5}$ was associated with a $2.24\%$ increase in the daily counts of confirmed cases. A study by Wu et al. (2020) indicates
that an increase of just $1\ \mu g\ m^{-3}$ in $PM_{2.5}$ is associated with an $8\%$ increase in the COVID-19 death rate ($95\%$ confidence
interval [CI]: $2\%$, $15\%$). Fattorini and Regoli (2020) showed that long-term air-quality data significantly correlated with cases
of COVID-19 in up to 71 Italian provinces, indicating that chronic exposure to atmospheric contamination may represent a
favourable context for the spread of the virus. That said, Contini and Costabile (2020) cautioned against translating high values
of conventional aerosol metrics, such as $PM_{2.5}$ and $PM_{10}$ concentrations to an increase in vulnerability or to a direct explanation
of the differences in mortality observed in different countries without chemical, physical, and biological analysis. Irrespective
of the consequences of elevated airborne PM concentrations, actions to mitigate the sources of that pollution rely critically on
knowing where and when their emissions occur.

Inverse modeling attempts to estimate on-the-ground emissions based on concentrations measured after the emissions have
been transported (Enting, 2002). The observations are linked to the emissions through the use of an atmospheric transport
model. This technique has been used to constrain greenhouse gas emissions estimates at global (e.g. Gurney et al., 2002;
Chevallier et al., 2010) and regional scales (e.g. Bréon et al., 2015; Lauvaux et al., 2016).

A small number of existing studies document earlier attempts to infer PM emissions maps from in situ PM measurements.
For example, Guo et al. (2018) evaluated estimates of $PM_{2.5}$ emissions in Xuzhou, China, using the same coupled Lagrangian
particle dispersion modeling system (FLEXPART-WRF) that is being used within the MAPM project. While similar in set-
up to MAPM, Guo et al. (2018) focus on a comparably much larger area, using a coarser-resolution transport model, and
substantially fewer measurement sites within the domain of interest. Furthermore, their study focused on predicting $PM_{2.5}$
concentrations for the purpose of forecasting rather than obtaining the best estimates of $PM_{2.5}$ emissions to identify source
regions. Application of their method demonstrated that inferred $PM_{2.5}$ emissions aggregated over Xuzhou ($11{,}258\ km^2$) were
$10\%$ higher than what was expected from a multi-scale emissions inventory. They identified that their inversion system could
be improved by increasing the number of sites at which $PM_{2.5}$ was measured and by reducing the uncertainty of the prior
emissions map.

The purpose of the MAPM (Mapping Air Pollution eMissions) project is to develop a new operational capability to generate
near real-time surface emissions maps of PM pollution as a service to city officials. Surface PM emissions maps are retrieved
from a combination of a prior (first-guess) emissions map, in situ atmospheric measurements of PM, and a description of air





parcel advection over the domain derived from a transport model driven by atmospheric wind fields. PM can be described by its 'aerodynamic equivalent diameter', and particles are generally subdivided according to their size: $<10$, $<2.5$, and $<1\,\mu m$ ($PM_{10}$, $PM_{2.5}$, and $PM_1$, respectively).

During the southern hemisphere winter of 2019, as part of the MAPM project, a field campaign was conducted in Christchurch, New Zealand to record in situ measurements of $PM_{10}$, $PM_{2.5}$, and $PM_1$ concentrations across the city (Dale et al., 2020b). These in situ measurements were taken in support of the inverse model. The winter season was selected as the time of the campaign because that is the time of year when aerosol emissions in the region are at their highest, due to the large amount of homes that use wood-burning heaters.

This study's main objective is to develop and test an urban inverse model for $PM_{2.5}$ emissions. This inverse model incorporates the in situ measurements recorded during the MAPM field campaign in conjunction with atmospheric transport models and an inventory-based first-guess estimate of emissions to create an optimized emissions estimate for the region. We first conduct and present a series of Observing System Simulation Experiments (OSSEs) that test the inverse model. We then showcase the results of the implementation of the model to the measurements obtained during the MAPM winter 2019 field campaign in
Christchurch.

## 2   MAPM measurement campaign

In this study, Christchurch was selected as a target city to demonstrate MAPM's capability, as Christchurch is New Zealand's third largest city (population of 385,500 as of June 2019) and is one of the most polluted cities in New Zealand, especially during winter. The main source of PM emissions in Christchurch in winter is burning wood and coal for home heating. Minor
anthropogenic sources of $PM_{2.5}$ are industry and transport, along with natural sources including dust and sea salt particles from the nearby ocean.

      Christchurch is the main urban centre of the Canterbury region, which is situated on the east coast on the South Island of New Zealand (see red box in Fig. 1). It is located on the eastern fringe of the Canterbury Plains, which slope gently from the coast to the Southern Alps that rise to elevations well above 3000 m. While Christchurch is situated on generally flat terrain, the
Port Hills, immediately south of the main urban area, form the northernmost side of the volcanic landscape of Banks Peninsula, and provide a local orographic feature that reaches elevations of up to 450 m (Fig. 1).

      The MAPM field campaign that provides the required PM concentrations measurements used in this study, took place from 21 June to 25 August 2019 in Christchurch and the immediate surrounding area. The locations of all instruments deployed during the field campaign are shown in Fig. 1. The field campaign, corresponding measurements and their uncertainties are
described in detail in Dale et al. (2020b). Briefly, two different types of instruments measuring $PM_{2.5}$ were deployed during the campaign: 17 ES-642 (Met One Instruments, OR, USA) remote dust monitors and 50 Outdoor Dust Information Nodes (ODINs) (NIWA, Auckland, NZ). ODINs are a low cost instruments that use the Plantower PMS 3003 sensor, which is described in Zheng et al. (2018). Both PM instruments are nephelometers, estimating the mass concentration of $PM_{2.5}$ based on the rate of scattering of laser light. For the ES-642, PM greater than 2.5 μm was filtered out with a sharp-cut cyclone filter and



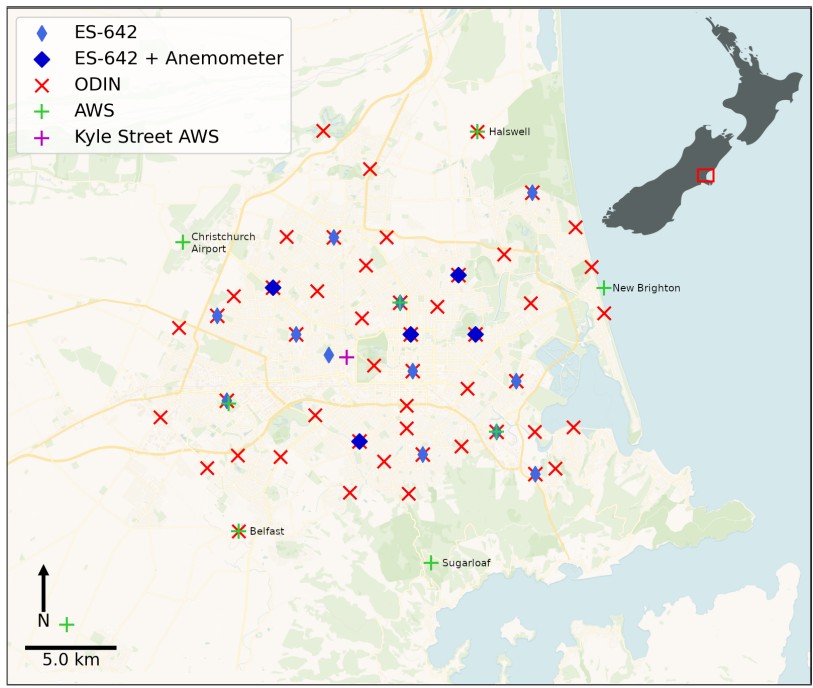

**Figure 1.** Map of Christchurch with the locations of the PM instruments and AWSs as deployed during the MAPM field campaign in 2019. The location of Christchurch within New Zealand is shown in the top right corner as indicated by the red box. © OpenStreetMap contributors 2020. Distributed under a Creative Commons BY-SA License.

the air coming into the sensor was heated to prevent water vapour being identified as PM. As ES-642s require mains power, they were installed in the backyards of residents, generally attached to fences or the sides of single story buildings. The installation height of the instrument was dependent on the location: the ES-642s needed to be connected to a power outlet, so their installation heights were dependent on what was available for mounting the instrument to; the ODINs, on the other hand, were mainly installed on light-poles unless they were co-located with the ES-642. Overall, the majority of the instruments (54 %)

were installed higher than 3 m above the ground, 20 % were installed below 3 m but above 2.5 m and 9 % above 2 m but below 2.5 m. Of the 50 ODINs that were deployed for the MAPM field campaign, 16 were co-located with the ES-642 instruments (one ES-642 site was deemed not suitable for a solar-powered ODIN) and the remaining instruments were installed throughout the city attached to light-posts (Dale et al., 2020b). Compared to the ES-642, the ODINs are much lower-cost, allowing for a larger network of instruments to be installed. While the ES-642s made instantaneous observations approximately every second,

the ODINs took a single instantaneous measurement every minute.

Immediately prior to and following the MAPM field campaign, all $PM_{2.5}$ instruments were co-located for one week. The co-location data were used to correct the $PM_{2.5}$ measurements from the ES-642s and ODINs against a reference instrument, the tapered element oscillating membrane (TEOM) instrument (Thermo Fisher Scientific, MA, USA). The TEOM is installed





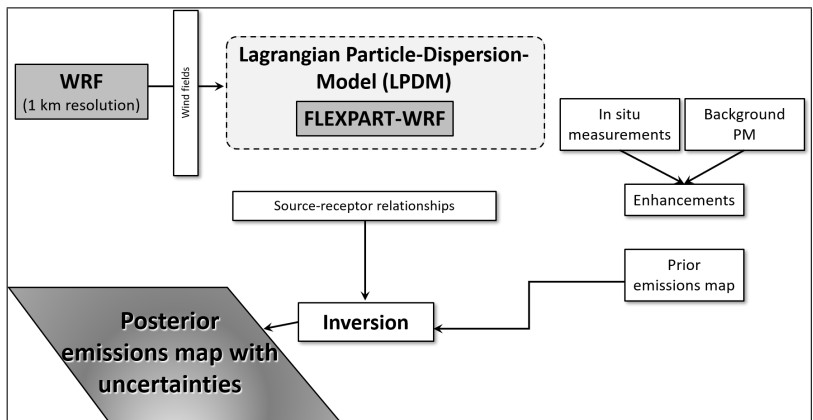

**Figure 2.** Overview of the inverse model set-up, its inputs and outputs as used and described in this study.

permanently at the co-location site, generating a consistent data set of PM measurements during the field campaign period. The
correction method applied is described in detail in Dale et al. (2020b).

In addition to PM measurements, meteorological conditions such as wind speed, wind direction, and temperature were
measured using several Automatic Weather Stations (AWS). Three AWSs were installed specifically for the MAPM campaign
on the outskirts of Christchurch (Fig. 1). These measurements were complemented by data provided from 13 AWSs that are
permanently installed and operated by the New Zealand MetService and the National Institute of Water and Atmospheric
Research (NIWA). Because the AWSs were operated by several different institutions, the variables recorded and the rate
that the data were recorded differ throughout the network. All AWSs measured wind speed and direction, temperature and
relative humidity, at a temporal sampling period ranging between 1 minute and 10 minutes – for consistency, all meteorological
measurements from all AWSs were averaged to a 10 minute temporal resolution.

## 3   Inverse modelling framework

As shown in Fig. 2, the inverse modelling framework derives its solution by combining several components. In general, the
measured concentrations are being optimized against the first–guess prior emissions map. However, the measured values of
interest need to be just the enhancements from the region of interest, so an appropriate background value representing con-
tributions from outside of the domain must be subtracted from the concentrations measured within the domain. The approach
to define the background can become complex. Additionally, in order for the measurements and the emissions estimates to be
compared, a transformation operator must be used to put them into the same unit-space. In this context, the transformation
operator relates the measurements to the emissions estimate through defining the source-receptor relationships at any measure-
ment site, as established through an atmospheric transport model used to define the potential source regions on the ground for
any particular measurement. Each of these component values is then integrated into the Bayesian inverse equation to calculate
an updated emissions estimate map. We expand upon the details of this procedure in the following sections.





As described above, given the meteorological input required to establish the source-receptor relationships, to estimate the PM emissions map, the inverse model must achieve a balance between the emphasis given to the prior emissions map and the in situ PM concentration measurements. This balance is achieved by prescribing adequate uncertainty to the prior and to the measurements. For a single inversion performed, i.e. a posterior emissions map derived for a single time step, where a prior emissions map and a series of measurements are provided as input, the quality of the posterior emissions map (driven by the derived uncertainties) is likely to depend heavily on the quality of the prior emissions map, as reported by Guo et al. (2018), and is consistent with previous outstanding literature (e.g. Gurney et al., 2005; Lauvaux et al., 2016).

## 3.1   Transport Modelling

In this study, the coupled LPDM FLEXPART–WRF model version 3.3.2 (Brioude et al., 2013) is used as the atmospheric transport that relates changes in emissions to changes in concentrations, the so-called source–receptor relationships (SRRs, Fig. 2). FLEXPART-WRF combines the Weather Forecast and Research Model (WRF) version 4.0 (Skamarock et al., 2019) and the FLEXPART model (Stohl et al., 2005; Pisso et al., 2019) version 9.

### 3.1.1   Weather Research and Forecasting Model – WRF

A challenge for modelling PM dispersion in an urban environment is accurately representing the meteorological conditions and capturing, with high fidelity, the influence of terrain under complex topography on the meteorology (Fay and Neunhäuserer, 2006). For the purposes of this study, we use WRF (Weather Research and Forecasting Model; Skamarock et al., 2005), a widely used numerical weather prediction model designed for operational weather forecasting as well as atmospheric research. Here, output from a WRF model simulation performed over the Christchurch domain at intermediate resolution (1 km; cf. 27, 9 and 3 km in Guo et al., 2018) is used to generate the required input to a Lagrangian particle dispersion model (LPDM).

The numerical weather prediction model WRF version 4.0 was employed to simulate the meteorological fields with four nested domains with horizontal resolutions of 27 km, 9 km, 3 km and 1 km and 38 vertical levels, as shown in Fig. 3. The boundary and initial conditions were initialized using the European Centre for Medium-Range Weather Forecasts (ECMWF) reanalysis data – ERA5 (Hersbach et al., 2020) with a horizontal resolution of 0.25°, 137 model levels at three-hourly time resolution. The WRF sea surface temperatures (SST) were initialized from the daily National Centers for Environmental Prediction (NCEP) high resolution SST analysis (0.083°). The CONUS physics suite (Liu et al., 2017) was selected for all the domains, which contains parameterisations for micro-physics, radiation, planetary boundary layer, surface layer, land surface and cumulus. Due to the high horizontal resolution of the final two domains (d03 and d04), the cumulus parameterisation was turned off for these domains. The standard high resolution static data were used for these simulations.

Overall a total of 32 WRF simulations were performed, covering the period from 21 June to 18 July 2019. Each simulation was initialized daily at 00:00 UTC and ran for a total of 72 hours, with the first 48 hours allocated for spin up, which was discarded from the final output files that were used in this study. Furthermore, only the meteorological output from domain 4 (d04), spanning the latitude range from 44.05°S to 43.25°S and longitude range from 172.29°E to 173.4°E, with a horizontal resolution of 1 km and a temporal resolution of 10 minutes was used as input to FLEXPART-WRF (Sect. 3.1.2).



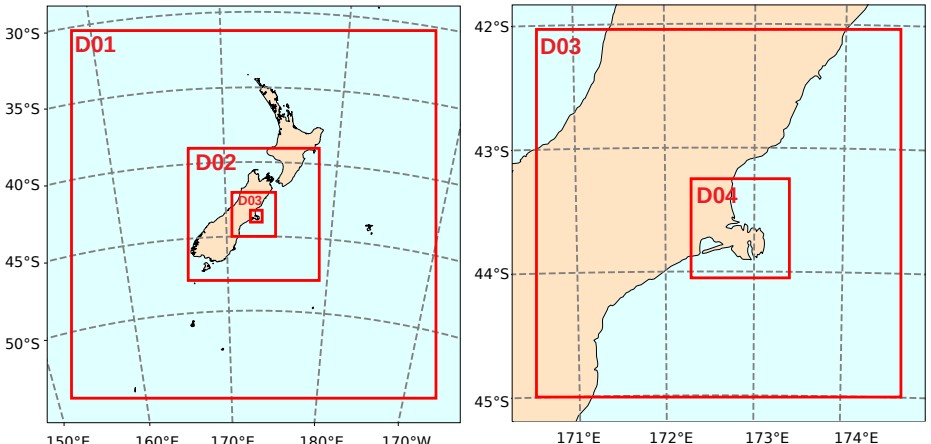

**Figure 3.** WRF domain configuration: (left) domains d01 and d02 with a horizontal resolution of 27,000 m and 9,000 m, respectively and (right) domains d03 and d04 with a horizontal resolution of 3,000 m and 1,000 m, respectively. Figure adapted from Lin et al. (2020).

**WRF evaluation against observations**

Temperature, wind speed, and direction, as simulated by WRF, were compared to measurements made using AWSs at seven different locations in and surrounding Christchurch. The 2 m temperature and 10 m wind speed from WRF were interpolated using a bi-linear interpolation to the latitude and longitude location of each AWS measurement site, forming a time series for comparison. The seven AWS sites, whose locations are marked in Fig. 1, measured wind speed at various heights ranging from 2 to 6 m. To make these measurements comparable to the 10 m wind that is provided by WRF, the measured wind speeds were

raised to 10 m using a log wind profile with a roughness length of $z_0 = 0.3$ m, which corresponds to a terrain with scattered obstacles.

    Modelled and measured temperature, wind speed and direction are shown in Fig. 4 for the period 28 June to 5 July 2019. During periods of low wind speed (e.g. 28–30 June) WRF accurately simulates the temperature during the day but overestimated the 2 m temperature by up to 9° at night. This bias is not apparent on other days when the wind speed is higher. This

suggests that WRF may have difficulty accurately simulating the temperature inversion layers that form in winter within the boundary layer, causing the surface temperatures to be overestimated by WRF. On warmer days and on days with higher wind speeds, WRF represents the measured temperature and wind direction at Kyle Street well, while it overestimates measured wind speeds. During the inversion period (22 June - 18 July) the difference between 10 minute mean AWS wind speed measured at Kyle St and the location-interpolated wind speed hindcast simulated by WRF was less than 1 ms$^{-1}$ 53 % of the time

and less than 2 ms$^{-1}$ 81 % of the time.

    The local topography may have an impact on the performance of WRF, as well. For example, the nighttime temperature biases for the downtown Kyle Street station, presented in Fig. 4, were the most extreme observed, and corresponding nighttime temperatures hindcast by WRF at several other AWS sites were also often several degrees warmer than was observed by





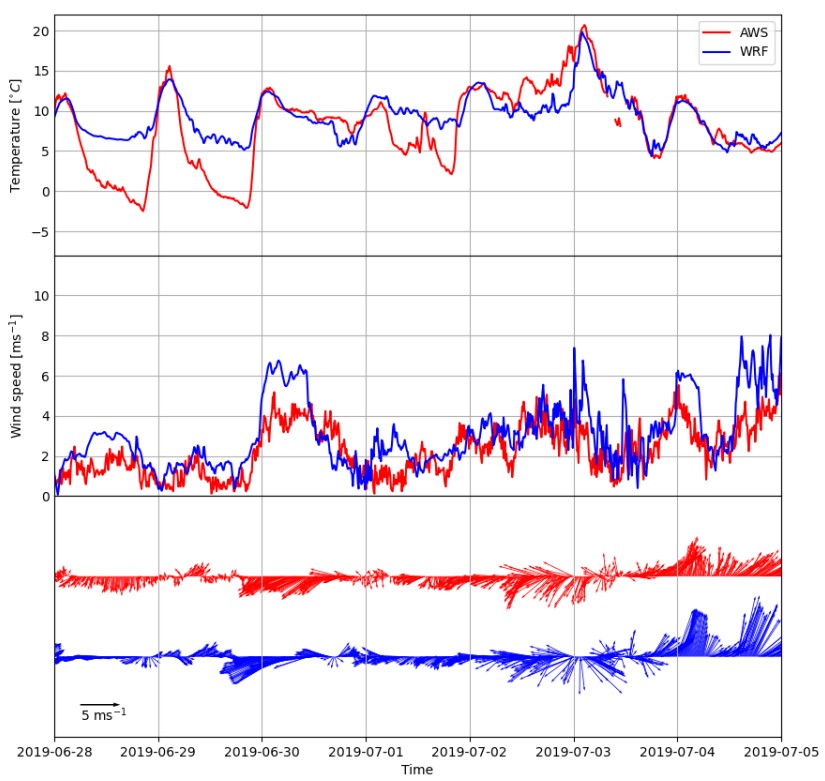

**Figure 4.** Example time series of 2 m temperature, 10 m wind speed and wind direction measured by the AWS installed at Kyle Street, compared to the temperature, wind speed and direction as simulated by WRF between 28 June and 5 July 2019.

the AWSs. However, this effect was not observed at the Mt Sugarloaf site which is located atop a peak in the port hills at

an altitude of 494 m. This is considered to be a consequence of the station height being above the height of many of the temperature inversion layers that form. Wind speed at the Sugarloaf site was also generally underestimated by a large margin. This is also unsurprising, as the wind flows at this site will be dependent on the surrounding complex topography that fall below the spatial resolution of the WRF model.

Aside from the specific areas of difficulty described, WRF generally captures the meteorological conditions of the region

within the typically-expected uncertainties. For example, the overestimation of the wind speeds near the surface in the WRF model are a known feature of the model (e.g. Shimada et al., 2011) and uncertainties remain on whether this bias is caused only under specific conditions. Overall, we have enough confidence in the WRF model to use it as the driver for the dispersion model in the inversion formulation, using an appropriate characterization of the relevant uncertainties.





### 3.1.2 FLEXPART-WRF and footprint calculations

Lagrangian Particle Dispersion Models (LPDMs) are widely used to compute trajectories of a large number of infinitesimally small air parcels (also referred to as particles) to describe the transport of air in the atmosphere. These models track the dispersion of a prescribed number of particles from their sources/sinks to designated receptors, i.e. measurement sites, when running forward in time, or from receptors to their sources/sinks when running backwards in time (Gentner et al., 2014). When particles are tracked backwards from a relatively small number of available atmospheric observation sites (i.e. receptors),

running LPDMs in backward mode is computationally more efficient than running the model forwards in time (Seibert and Frank, 2004). While LPDMs are widely used in atmospheric inversion studies for estimating regional fluxes, i.e. emissions (Maksyutov et al., 2020), there are not many studies that have used LPDMs to infer air pollution sources at city scales. Although not used to infer air pollution sources, Trini Castelli et al. (2018) demonstrated the capabilities of a 3-dimensional LPDM driven by 3-dimensional flow and turbulence input in both idealized and realistic urban mock-ups. Gariazzo et al. (2007)

used the SPRAY (Tinarelli et al., 1994) LPDM to evaluate the relative impact on air quality of harbour emissions, with respect to other emission sources located in the same area, for the city of Taranto, Italy. Rotach (2001) used an LPDM to compare an 'urban'-scale simulation in which the roughness sub-layer's turbulent structure is taken into account against a 'non-urban' simulation in which it is not. They found that neglecting the roughness sub-layer resulted in the largest errors for sources close to the surface and under conditions of mechanically dominated turbulence, highlighting the challenges of accurately simulating

air parcel trajectories through urban domains.

Three available LPDMs that have been widely and frequently used to model atmospheric transport processes include the Hybrid Single-Particle Lagrangian Integrated Trajectory (HYSPLIT; Stein et al., 2016) model, the Stochastic Time-Inverted Lagrangian Transport (STILT; Lin et al., 2003) model, and the FLEXPART model (Stohl et al., 1998, 2005). Hegarty et al. (2013) showed that all three models had comparable skill in simulating the tracer plumes when driven with modern meteoro-

logical inputs (such as WRF), indicating that differences in their formulations play a secondary role. In this study, to derive the source-receptor relationships that will be used in the inversion model framework, we use the FLEXPART model, which is widely used and has been extensively validated (Castro et al., 2012) . FLEXPART was originally developed to calculate long-range and mesoscale dispersion of air pollutants from point sources, such as those occurring after an accident in a nuclear power plant (Stohl et al., 2005; Pisso et al., 2019). Over the last decades, FLEXPART has been further developed and has

evolved to be a comprehensive tool for atmospheric transport modelling, attracting a global user community. The required meteorological input to FLEXPART is provided by the WRF model. The capability of FLEXPART to use WRF output was developed by Brioude et al. (2013) and this specific version of FLEXPART is referred to as FLEXPART-WRF, which is the version of FLEXPART used in this study.

Using the meteorological output at a horizontal resolution of 1 km from WRF (Sect. 3.1.1) as input to FLEXPART-WRF,

FLEXPART-WRF was used to derive the source-receptor relationships (SSR) by running the LPDM backward in time, e.g. particles were released from a measurement location to identify potential upwind sources. As the current version of FLEXPART-



WRF does not support the direct simulation of PM$_{2.5}$ concentrations, the FLEXPART-WRF simulation was set up to simulate the particles as passive tracer air parcels without considering chemical reactions and dry or wet deposition.

To establish the SSR for each hourly mean PM measurement, FLEXPART-WRF simulations were initiated at the end of the one hour period (noting that FLEXPART-WRF is running backward in time). Then, during the one hour period, 10,000 particles were released continuously from 49 receptor locations that correspond to the measurement sites described in Sect. 2 and traced backwards in time over a 24 hour period to establish their potential sources. The release altitude was set to 2 m for all releases and the FLEXPART-WRF output was saved every 30 minutes on the same grid as the WRF output was provided, i.e. same domain with a 1 km horizontal resolution. Overall, FLEXPART-WRF was run independently for each hour of the 31 days spanning the period from 22 June to 18 July 2019.

FLEXPART-WRF output, provided in units of residence time [seconds], can be used to identify the origin of emissions sources that contribute to the concentrations 'measured' at the receptor. We use the FLEXPART-WRF output to create 'footprints' that describe the backwards-in-time local contributions to any in situ measurement by integrating over 12 hours of output and over a 60 m thick surface layer. Twelve hours was chosen to be enough time for the particles to traverse the domain even under weak wind conditions, and 60 m was chosen as the integration height to be high enough to capture any elevated emissions sources. These footprints define the transport matrix (**H**), which is used to relate the emissions in flux-space to the measurements in concentration-space, as shown later in Eq. 1.

### 3.2 Prior PM$_{2.5}$ emissions estimates

Derived bottom-up emissions estimates used in this study include PM$_{2.5}$ emissions sources from: (i) traffic, (ii) industry (the top 13 industries only) and (iii) home heating. Across the whole domain, home heating contribute 72 % to the total PM$_{2.5}$ emissions on a typical winter day, while traffic and industry emissions only contribute 20 % and 8 % to the total emissions, respectively. The relatively high amount of emissions coming from home heating was a primary driver for the measurement campaign being focused in the wintertime.

For this study, Environment Canterbury (ECan) provided an estimate for the average PM$_{2.5}$ emission of a wood burner based on (a) 2018 census data, (b) issued permits for wood-burners, and (c) the estimated emissions per burner type. It should be noted that ECan formulates their estimate with an eye towards the worst-case scenario, as this would be most relevant to negative human health impacts, and thus they should be expected to overestimate emissions.

The 2018 census data were compiled based on a set of questions taken to the Christchurch community regarding whether or not they used wood for home heating. The census data were provided on statistical areas, which are irregularly-shaped polygons of various sizes covering the Christchurch city domain. To estimate the PM emissions from households in Christchurch, the number of households using a wood burner was multiplied by the average PM$_{2.5}$ emissions per wood burner. Of these total household emissions, 75 % were then used as an estimate for PM$_{2.5}$ emissions on a typical winter night. Here the assumption is that the type of wood burners and the percentage of active burners on a given night remain consistent across the city. The industry emissions for the top 13 emitters were calculated based on discharge estimates from ECan, and together they represent around 77 % of estimated industrial PM$_{2.5}$ emissions for the Christchurch air-shed. Vehicle emissions were also provided by





ECan and are based on 2014 data from the Christchurch Transport Model and emission factors were derived from the vehicle emissions prediction model that was developed by the New Zealand Transport Agency.

To derive hourly estimates of $PM_{2.5}$ emissions, the total 24-hour emissions were divided according to the estimated contri­bution of that hour to the total daily emissions for each of the three sources separately - these estimates were provided by ECan

for home heating and traffic. Due to a lack of any detailed information about the timing of the emissions released by industries, it was assumed that the industry emissions were released evenly throughout the 24-hour period. Overall the highest emissions occurred during nighttime when people were at home and starting their fires.

The household-based emissions per statistical area and the traffic emissions were rasterized onto a 10m by 10m New Zealand Transverse Mercator 2000 (NZTM2000) grid. Industry emissions were added as point sources on this grid. During the rasteri­-

zation, the conversion from grams per meshblock (statistical area) to grams per square metres was calculated. The NZTM2000 grid was interpolated and projected onto the 4[th] WRF domain (see below for more details, Fig. 3), using conservative interpo­lation which preserves the total emissions. The bottom-up estimate for $PM_{2.5}$ emissions in Christchurch over a 24 hour period is shown in Fig. 5, representing emissions on a typical winter day.

The hourly-resolution maps are used to construct the final prior emissions map that is used in the Bayesian inversion equation.

First, for any given hour with a given set of hourly measurements, the corresponding emissions map is a mean emissions map across the preceding 12 hours of emissions. This averaging is done to match the 12-hour length of the footprints, so that it is appropriately representative. Then, the definitive prior emissions map is a mean of each of these hourly emissions maps across every hour in the inversion period.

### 3.3  Determination of the Background Concentrations

Since we are only concerned with constraining the emissions coming from a local region (here within Christchurch), the measurements being used in the inverse model should similarly also only reflect the enhancements that have come from that region, excluding any contribution coming from outside the city. As a result, any potential influx of PM into the domain (here referred to as the background) needs to be subtracted off each measurement taken within the city domain, leaving only the locally-measured 'enhancement' values.

Defining the background air mass is often one of the most difficult, yet important challenges during the inversion formulation (e.g. Göckede et al., 2010; Lauvaux et al., 2016). In the $CO_2$ community, for example, this is often achieved by selecting a measurement site to act as a background based on wind direction (e.g. Kort et al., 2012; McKain et al., 2012; Lauvaux et al., 2016) - and a similar approach is applied here. During the measurement campaign, to estimate the inflow of PM into the domain, a number of PM sensors were deployed at the perimeter of the city. Measurements from these 'background' sites were

used, together with wind direction measurements, to estimate the background PM concentrations flowing into the city.

Specifically, a total of five AWS/ODIN pairs were selected along the perimeter of Christchurch and used to determine the background PM concentrations. The five sites-pairs are specifically marked in Fig. 1, where we included the name of the site location. To determine whether or not the wind is blowing into the domain, Hagley Park was chosen to be the center point of the region, and the angle between the location of the AWS and the park was estimated - which corresponds to a wind direction





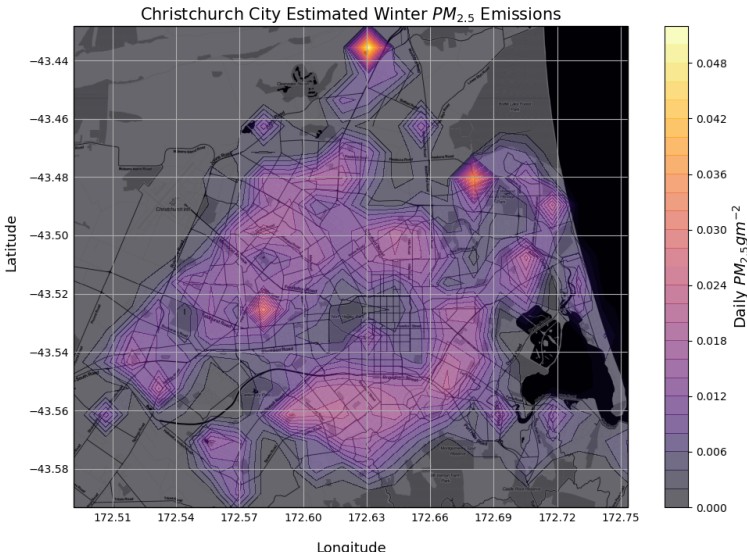

**Figure 5.** Bottom-up emissions map over Christchurch, interpolated onto the 1 km WRF grid and used as the prior for the inversion calculation. PM$_{2.5}$ emissions sources include: (i) household heating, (ii) traffic, and (iii) industry. Contour lines are shown per every 0.002 g m$^{-2}$ of PM$_{2.5}$

.

marking background PM flowing into the domain. The calculation of the background concentration value at any given hour then used the following procedure:

1. For each AWS site, a range of wind direction angles are determined that correspond to the wind flowing into the domain, viz:

   – Halswell AWS site: wind directions need to be at $22.5° \pm 22.5°$,

– New Brighton AWS site: wind directions need to be at $67.5° \pm 22.5°$,

   – Sugarloaf AWS site: wind directions need to be at $168.75° \pm 22.5°$,

   – Belfast AWS site: wind directions need to be at $225° \pm 22.5°$,

   – Christchurch Airport AWS site: wind directions need to be at $303.75° \pm 22.5°$,

2. Using the one-minute-resolution data for the ODIN corresponding to the AWS site and the respective five-minute-
resolution wind data, all the PM$_{2.5}$ measurements that correspond to the air flowing into the domain are selected. This results in a collection of PM$_{2.5}$ measurements that were made when the wind direction pointed into the domain for each background site.





3. For each AWS/ODIN pair, the $PM_{2.5}$ measurements that had been selected in the previous step were then pre-screened for local influences based on (i) if the change in $PM_{2.5}$ from one minute to the next is greater than $50\,\mu g\,m^{-3}$, (ii) if they exceed $2\times$ the calculated hourly standard deviation.

4. All surviving $PM_{2.5}$ measurements from all sites are combined into one time series, and bin them into hourly averages.

5. A 10-day rolling median of the hourly time series is computed, to remove any remaining local influences. This rolling median value is then used as the background across the whole domain at a given hour of the day.

### 3.4 Bayesian Inversion Calculation

Given the prior flux estimates of the emissions and the in situ concentration measurements, the optimal emissions flux map is computed using the Bayesian inversion calculation, following the formula in Tarantola (2005):

$$\mathbf{x} = \mathbf{x_0} + \mathbf{B}\mathbf{H}^T(\mathbf{H}\mathbf{B}\mathbf{H}^T + \mathbf{R})^{-1}(\mathbf{y} - \mathbf{H}\mathbf{x_0}). \tag{1}$$

Here, $\mathbf{x}$ is the posterior flux map, $\mathbf{x_0}$ is the prior flux map, $\mathbf{B}$ is the prior error covariance matrix, $\mathbf{H}$ is the influence function (more specifically the transport matrix, containing the 12-hour footprints for each hourly measurement), $\mathbf{R}$ is the error covariance matrix for the observations, and $\mathbf{y}$ is the 1-dimensional vector of hourly-averaged observations from the region (or enhancements in cases where there is background influence that must be removed before their use in the inversion).

The posterior error matrix is defined as:

$$\mathbf{A}^{-1} = \mathbf{B}^{-1} + \mathbf{H}^T\mathbf{R}^{-1}\mathbf{H}. \tag{2}$$

The errors on the prior flux estimates are assigned in a manner similar to what has been described in Nathan et al. (2018) and Lauvaux et al. (2016). The error covariance matrix for the prior fluxes, $\mathbf{B}$, contains the variances for every grid-point in the domain along the diagonal, and the cross-correlation terms in the off-diagonals. The variances are defined as the square of the root mean square (RMS) values at each grid-point of the flux map, and the RMS values are set to be 50% of the corresponding flux value. The covariances, representing the spatial error correlations, are then defined using an exponential decay function with a 1-km correlation length. Here, we have set the correlation length to match the grid-cell size in the domain, to allow for more independence in the allocation of emissions adjustments during the inversion process. However, even with this near-independence, we still maintain some spatial error correlations with the understanding that they are necessary to keep the inverse problem regularised, and thus to prevent a divergence of the solution (Bocquet, 2005). For the error covariance matrix for the observations, the variances are defined as the square of the measurement uncertainties that are provided with the data sets (Dale et al., 2020b). The off-diagonal elements of the covariance matrix are all set to zero; i.e. all measurements at different locations and times are considered to be independent from one another.


## 4   Observing system simulation experiments – OSSEs

Observing system simulation experiments (OSSEs) are controlled experiments to test the sensitivities of an observing system to changes in its set-up (Masutani et al., 2010). These can be especially valuable for gaining insight into complex systems, such as the inversion framework, which has many degrees of freedom. We undertake several OSSEs here to investigate multiple

observational network designs and their impact on inferred emissions maps as shown below. In general, the OSSEs will follow the same inversion schematic of Fig. 2, except that no background value will need to be subtracted off, as, by design, all 'measured' emissions will have originated from within the domain of interest.

### 4.1   Sensitivity of the inversion system to measurement uncertainty

To investigate the sensitivity of the inversion system to varying measurement uncertainties, we performed a series of OSSEs. A

series of eight different sensitivity experiments were conducted. These experiments have identical set-ups and input parameters besides the change in the magnitude of the measurement uncertainties. Each of these trials was performed on a daily-inversion basis, with 10 inversions being calculated over the course of 10 days: 23 June to 2 July, 2019. The flux error characterizations follow the description in Sect. 3.4.

For all experiments the procedure involves the following steps:

– Generate a set of 'true' $PM_{2.5}$ concentration measurements at each of the measurement locations by multiplying the transport matrix $\mathbf{H}$ (i.e. the generated 'footprint' at every site at every hour) by the corresponding true flux map. Here we use the prior emissions map (Sect. 3.2) as the true flux map. This idealised set-up ensures that no transport error is introduced into the system.

– Add noise to the 'true' concentration time series to generate a synthetic $PM_{2.5}$ concentration measurements time series.

The noise on the measurements is sampled from a normalized Gaussian distribution with a mean of 0 and a standard deviation of the RMS value of the measurements. The magnitude of the RMS value is the variable that gets modified in the eight different OSSE experiments.

– Add noise to the true flux to create the prior flux required for the inversion. This noise is 'colored', i.e. taken from the correlation matrix that was used in the construction of the $\mathbf{B}$ matrix, to ensure that the prior flux is including influences

from every spatial correlation. This is done only after the correlation matrix has been confirmed to be positive definite, or else the nearest positive definite matrix is computed and used instead.

– Calculate a posterior emissions map and its uncertainties by using the Bayesian inversion equation described in Sect. 3.4, taking the synthetic measurement time series, the prior emissions map, and each of their respective uncertainties as input.

– Compare the derived daily posterior emissions map against the true emissions map to assess the success and sensitivities of the inverse model.



**Table 1.** Summary of the set-up for all eight experiments performed in this study.

| Experiment # | Total # of sites included in Experiment | RMS scaling applied to 18 sites below/above 10 mg m3 | RMS scaling applied to 31 sites below/above 10 mg m3 | Gain ± standard deviation |
|---|---|---|---|---|
| 1 | 49 | 10% / 10/% | 10% / 10% | $0.4246 \pm 0.0594$ |
| 2 | 49 | 90% / 90/% | 90% / 90% | $0.1591 \pm 0.0413$ |
| 3 | 49 | 10% / 10/% | 90% / 90% | $0.3180 \pm 0.0556$ |
| 4 | 49 | 90% / 90/% | 10% / 10% | $0.3749 \pm 0.0652$ |
| 5 | 49 | 90% / 10/% | 10% / 90% | $0.3415 \pm 0.0447$ |
| 6 | 18 | 10% / 90/% | 90% / 10% | $0.3549 \pm 0.0534$ |
| 7 | 31 | N/A | 10% / 10% | $0.2918 \pm 0.0525$ |
| 8 | 31 | 90% / 90/% | N/A | $0.1204 \pm 0.0489$ |

Table 1 shows the experimental set-up for the eight OSSEs performed. The values of 90 % and 10 % were intentionally chosen as extremely high and low, respectively, so that the OSSEs test the limitations of the system. Experiments 1 and 2 establish upper and lower bounds with respect to the measurement uncertainty definitions on the system. Experiments 3 and
4 examine the effects of having starkly different measurement uncertainties at 18 of the 49 measurement sites. Experiments 5 and 6 explore the effects of having measurement uncertainties that change characteristics above a $10 \, \mu\text{g} \, \text{m}^{-3}$ threshold on the same partitioning of sites. Finally, Experiments 7 and 8 look directly at how having fewer instrument sites (18) with lower uncertainty compares to having more instrument sites (31) with higher uncertainty.

Two metrics, gain (G) and error reduction (ER), are used following Lauvaux and Davis (2014) to quantitatively evaluate the
inversion performance during the OSSEs. The first of these metrics is the global gain, which is a measure of improvement of the posterior flux over the prior flux relative to the true flux:

$$\text{G} = 1 - \left( \frac{\sum\limits_{i=1}^{N} |x_{\text{true},i} - x_{\text{post},i}|}{\sum\limits_{i=1}^{N} |x_{\text{true},i} - x_{\text{prior},i}|} \right) \tag{3}$$

where $i$ keeps track of the individual grid-cells across the total number in the domain $N$, $x_{\text{true}}$ is the true flux, $x_{\text{post}}$ is the posterior flux, and $x_{\text{prior}}$ is the prior flux. The second metric is a grid-point-by-grid-point comparison of the error reduction,
expressed as a percentage difference between the posterior uncertainty values from the diagonal of **A** compared to those from the diagonal of the prior uncertainty matrix **B**:

$$\text{ER} = 1 - \left( \frac{\text{diag}(\mathbf{A})}{\text{diag}(\mathbf{B})} \right). \tag{4}$$

For all experiments, daily posterior emissions maps (i.e. 10 per experiment) were computed using the synthetic measurement time series, their footprints, and the prior emissions map. For each experiment, we calculated the overall gain and error
reduction, before the mean gain value and its standard deviation was calculated across the 10-day period. These mean gain values and their standard deviations are presented in Table 1.





The results from Experiments 1 and 2, intended to set baselines for all-low and all-high uncertainties, respectively, show that having a lower uncertainty of 10 % on all measurements from all sites yields a much higher average gain, and vice versa. Both Experiments 3 and 4 have a relatively high gain, if Experiment 1 is used as a reference for what 'high' gain looks

like. Experiment 3 results in a gain value that is about 0.06 lower than the mean gain derived from Experiment 4, noting that experiment 3 had 18 sites with low uncertainties while Experiment 4 had 31 sites with low uncertainties assigned to the synthetic measurements. This seems to indicate that a greater number of measurement sites with low uncertainty will lead to an increase in overall gain. However, the converse should also be acknowledged, that in Experiment 3, the number of measurements with a larger uncertainty, i.e. 90 % was higher than in Experiment 4, 18 versus 31 sites. As a final point, it is

noteworthy that even in Experiment 3, where only 18 sites out of the total of 49 have a low uncertainty associated with them, the gain value indicates that the inversion system works well, and it is comparable to that of Experiment 1, where all 49 sites were associated with a low uncertainty.

The results from Experiments 5 and 6 suggest that, in a scenario where the uncertainty is dependent on the measured concentration, there is little impact on the gain, compared to Experiments 3 and 4, for example. The obtained gain from

Experiments 5 and 6 lie between the gain values from Experiments 3 and 4. This results, in part, from the fact that just over half of the measurements were above the chosen threshold, i.e. 57 % of all measurements in Experiment 5 and 61 % in Experiment 6. Note that the percentages are not identical because of the random nature of the noise that is added to the true measurements to create the synthetic data used in the inversion. The results of the experiments add further credence to the idea that increasing the amount of low-uncertainty measurement sites drives the gain much stronger than increasing the number of

high-uncertainty sites.

Experiments 7 and 8 address more directly the question of whether a low number of sensors with a small uncertainty can perform as well as a high number of sensors with a high uncertainty. Similar to what was surmised from the results of the preceding experiments, here it becomes more definitive that a low number (18) of sensors with a small uncertainty outperforms the higher number (31) of sensors having a high uncertainty. While 18 measurement sites is still a very high number compared

to many $CO_2$ inversion studies, for example, the important conclusion here is that the information gained from having small uncertainties associated with a lower number of sensors cannot be compensated for by a relatively large increase in the number of sensors with a high uncertainty.

Figure 6 showcases the mean error reduction maps associated with each of the 8 experiments. Error reduction maps are generally useful for identifying spatially where the improvements in the model are or are not being attributed. Here we see that

such improvements are generally restricted to the city itself, with more error reduction in the areas with more measurement sites. This is generally expected, as a greater amount of measurement sites increases the amount of information about a region (particularly as footprints start to overlap), allowing the system to assign much less uncertainty to those areas. It follows that the large number of measurement sites in this experimental set-up, generally, means that the only areas with a low amount of error reduction are around the outskirts of the city, where footprints from measurement sites are less likely to overlap. Further,

the relatively large number of sites means that the differences in spatial features between experiments is often subtle.



**Figure 6.** Error reduction maps across the eight different experiments. These are expressed as a percent difference between the posterior uncertainty and the prior uncertainty at any grid-point. Each map is a mean across the 10 single-day calculated inversions. Red circles indicate instrument site locations.





Looking at the error reduction patterns of the city as a whole for each experiment shows that they seem to follow the same trend as was seen in the gain calculations. The higher the gain, the larger the amount of error reduction, generally speaking. Subtle differences, e.g. between Experiments 3 and 4, are difficult to identify, and may not lead to generalizable conclusions. Broadly speaking, these maps seem to support the conclusions drawn from the gain analysis that experiments with

a larger number of sensors with low uncertainty tend to yield better results. Further, the difference in error reduction between Experiment 7, with a low number of low-uncertainty sensors, and Experiment 8, with a high number of high-uncertainty sensors, makes the improvement of the model between those experiments more apparent.

## 4.2    Testing a time-evolving prior

Two additional OSSEs were designed to investigate an approach where inversions could be used in an incremental, time-

evolving fashion to improve the prior emissions estimates from day $n$ to day $n+1$. The motivation for this was to test if this would be a viable method for deriving a near-true prior flux map for inversion investigations in regions where either no prior emissions estimates exist or perhaps only a very poor estimation in need of substantial improvement. The ability to obtain a prior flux estimate with a very low associated uncertainty would be extremely useful for monitoring local emissions.

An OSSE framework was first designed to be very similar to the one described in Sect. 4.1. The same domain with a

horizontal resolution of 1-km is used, as well as the footprints that were calculated for all 49 measurement site locations. The uncertainties on the fluxes, including spatial correlations, are again characterized following the description in Sect. 3.4. The RMS value on the measurements is set to be 10 % of the measurement value, and the corresponding noise on the synthetic measurements is sampled from a normalized Gaussian distribution with a mean of 0 and a standard deviation of the RMS value. For both OSSEs, synthetic measurements were created for the entire 10 day period from 23 June to 2 July 2019, using

the same 'true' flux emission map as the OSSEs described in Sect. 4.1. Here, the uncertainty calculations of the **B** and **R** matrices from Eq. 1 are calculated for each inversion separately, however, if this methodology is shown to be viable, then a reassessment of the uncertainty calculations will need to be undertaken, which accounts for the fact that each day's inversion is no longer independent from the preceding days.

The OSSEs performed here are broken down into 10 consecutive 1-day inversion calculations. We aim to test whether

an incorrect prior emissions map can be improved progressively over time with the information from the measurements and corresponding footprints. Thus, the method being tested involves using the posterior emissions map computed for the period of day $n$ to be used as the prior emissions map for day $n+1$. These OSSEs are performed to test whether the additional information provided by the measurements can lead to an improved prior emission map over time, perhaps even detecting areas that had been misidentified in the original inventory estimates.

The two experiments that were designed to test this approach have their prior emissions maps on day 1 defined as follows:

- A – The prior emissions map on day 1 is scaled up uniformly to have 1.2 times the total amount of emissions coming from the 'true flux' map.

- B – The prior emissions map on day 1 has one high-emission area from the true emissions map set artificially low.





In both OSSEs, we test whether the true flux map can be retrieved after 10 days of time-evolving iterations on the prior.
Figure 7 shows the true flux map that was used together with the transport matrix **H** to create the measurement time series
for the period as well as the corresponding, incorrect, prior flux estimate maps for day 1 of each OSSE scenario. For Scenario
A, the 20 % increase across every pixel is most evident in the colour-map among the grid-cells showing high emissions. For
scenario B, an 8-grid-cell region near the south-central part of the domain has had their values reset to 50 $\mu g\ m^{-3}$, which is
very low compared to the true flux emissions.

**Figure 7.** The true flux map (top left), which was used to create the measurements time series during the period of interest, and the corresponding day 1 prior emissions estimates for OSSE Scenarios A (top right) and B (bottom).

The two OSSEs testing the idea of whether a time-evolving prior map leads to a continuous improvement of the prior
emissions were inconclusive due to some apparent feedback-type process that causes the posterior map to create compounding



artifacts over time, as evidenced in Figure 8. Figure 8 shows the difference between the posterior map and the true flux map after day 1 and after day 10 for Scenarios A and B, respectively.

**Figure 8.** Difference maps between the posterior flux and the true flux after inversions on days 1 (left) and 10 (right) of OSSE Scenario A (top row) and Scenario B (bottom row).

The maps on day 1 in both scenarios showcase results from a normal-style inversion, since they only incorporate the prior from day 1. The maps on day 10 show how those results have changed after chaining together 10 days of posteriors into priors. Starting with Scenario A (top row in Fig. 8), while there were areas of disagreement after day 1, particularly in the south-central portion of the domain, the difference map on day 10 has amplified differences in other parts of the map that previously did not exist, particularly in the west and north of the domain. A comparison of the maps across each day does show a gradual



decline over time between days 1 and 10, implying a potential feedback mechanism. A similar effect can be seen in the maps
for Scenario B. Here, the inversion after day 1 did not do a particularly strong job of recovering the 8 artificially-low pixels
in the south-central part of the domain. The estimates for these pixels are noticeably improved by day 10. However, as with
Scenario A, other parts of the domain that previously had low differences between the posterior and truth have gotten worse
over time, particularly in the western and northern parts of the domain. In both scenarios, even in spite of areas that do improve
over the course of 10 days, there is some apparent feedback mechanism in other parts of the map that pulls the solution farther
away from the truth over time.

The initial hypothesis for this feedback loop is that grid-cells that are semi-arbitrarily assigned higher values after day
1 will be assigned a larger uncertainty going into the future days, which increases the chances of being attributed higher
fluxes, etc. This hypothesis was tested by rerunning the analysis while maintaining a static flux uncertainty matrix and flux
noise vector across all of the days. This analysis showed the hypothesis to not be sufficient to explain the observed behavior.
Future investigations into this methodology may want to focus on testing whether adjustments made to the cost function of
the inversion equation itself can delineate and ideally eliminate the root cause of the apparent feedback loop. Additionally, the
influence from a dynamic meteorology during the period contributes in some way to the inconclusive results of these OSSE's.

## 5 Inversion calculation using measurements obtained during the MAPM campaign

An inversion calculation was performed using the measurements obtained during the MAPM field campaign (Sect. 2, hereafter
referred to as the real-data inversion to distinguish from the OSSEs). The period for the inversion was chosen to be from 22
June 2019, 12:00 UTC to 18 July 2019, 24:00 UTC. The inversion was then run for the 'daytime' and 'nighttime' separately,
where daytime was defined to go from 6am to 6pm local time (18:00–6:00 UTC, excluding 6:00 UTC) and nighttime to go
from 6pm to 6am local time (6:00–18:00 UTC, excluding 18:00 UTC). One posterior map per day was computed for each
daytime or nighttime period.
The inversion system components are shown in Fig. 2 (Sect. 3). Where the real-data analysis differs from the OSSE inves-
tigations in set-up is (i) in the necessitation of defining background values, following the description in Sect. 3.3; (ii) in the
need to filter measurements due to rain (see below); and (iii) in the exclusion of the step wherein noise would be added to
the flux or observations, as had been done for the OSSEs, since the prior flux and the observations now have their own errors
built-in. Additionally, the corresponding results will require a slightly different analysis approach, since the 'truth' is not a
known quantity.

Because rain is known to wash out particulate matter from the atmosphere (e.g. Atlas and Giam, 1988), which would lead to
artificially low measurements and therefore would bias the inversion results, all hours during which rain occurred were removed
from the data set. The 12 hours following any rain event were also removed from the data set, to prevent over-counting in our
inversion setup which uses 12-hour footprints, i.e. this allows the emissions to "build back up" enough in the atmosphere to
match the 12-hour integrated footprints that are used in this study. The 'rainy hours' were identified by using data from the
AWS that is installed close to the city centre, at Kyle Street (purple cross in Fig. 1), wherein any hour that recorded any nonzero




precipitation value was deemed to indicate a potential rainfall event in the area during that hour. The PM$_{2.5}$ measurements at that hour and the following 12 hours for all PM measurements at all sites were thus excluded from the analysis.

The results of the real-data inversion indicate a significant overestimation in the inventory estimates for PM$_{2.5}$ emissions in Christchurch. Figure 9 shows the time series comparison for prior-based flux estimates compared against the posterior flux estimates for the daytime and nighttime, respectively. The values shown are sums across the entire domain, for either the prior or posterior maps. The prior map at any hour is a mean of the preceding 12 hourly estimated emissions maps, a length that was chosen to correspond to the 12-hour footprints for the measurements at that hour. A scale factor is derived for the creation of the corresponding "hourly" time series of posterior values, since only one posterior map is computed per daytime or nighttime period. This scale factor is defined as the proportion of the domain sum of the posterior emissions map divided by the domain sum of the mean prior emissions map (across all included hours) used in the inversion for the given daytime or nighttime period.

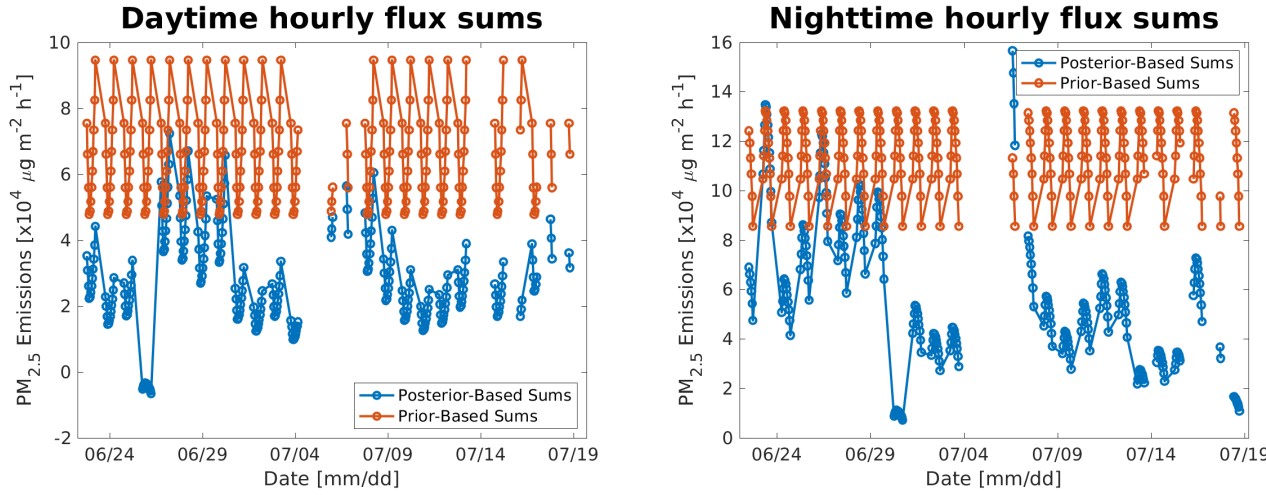

**Figure 9.** Time series of sum of the fluxes across the domain in the prior vs. posterior for the daytime and nighttime inversions.

For both daytime and nighttime, the results indicate a 40 to 60 % overestimation of the prior emissions compared to the posterior. This is believed to be, in part, an expected result following several assumptions that had been made when calculating the PM$_{2.5}$ inventory (Tim Mallett from ECan, personal communication, October 2020). These assumptions include:

– That 80 % of all wood burners present in Christchurch are operating on a typical winter day, which will be an overestimate, especially on days with higher temperatures.

– The fuel use estimates assume around 10 hours of use per day for wood burners. This can be an overestimation as (i) new homes have more insulation than previous houses, (ii) heat-pumps are installed in new homes rather than wood-burners, which lead to no PM emissions, and (iii) a smaller floor area or fewer occupants would reduce the burning time.





- That all wood burners over $1.0$ g kg$^{-1}$ are combined into one emission factor category, though a future version of the inventory separates out $1.0$–$1.5$ g kg$^{-1}$ from those over $1.5$ g kg$^{-1}$, which could further lower emissions estimates.

- Ambiguity in the underlying census data used to tally the number of active burners or the time they are used for, which only asks a homeowner if they use a wood burner, but does not include a question around whether or not they have an alternative heating method or for an estimate of the average amount of time that they use the heating.

As a result, the inventory comes with a large uncertainty that favours the possibility of an overestimation. Thus, the results of the inverse model correcting that down, i.e. to lower emissions as in Fig. 9, can be viewed as sensible. Recent OSSEs by Lauvaux et al. (2020) indicate that an inversion set-up like the one presented here should be able to recover values close to the true emissions if the prior fluxes are offset by as much as $20\%$, however large offsets above $40\%$ may not be able to be fully recovered, and may still require an extra correction (in their study, a $40\%$ offset still was $18\%$ away from the truth after inversion). This may imply that the apparent overestimation is still larger than our posterior results indicate.

The nighttime period is traditionally left out of inversion analyses because of difficulties in the model to accurately capture the nocturnal boundary layer (Geels et al., 2007; Steeneveld et al., 2008). For example, Lac et al. (2013) found a positive bias of 5 ppmv of $CO_2$ in urban and suburban regions of Paris attributed just to boundary layer height error in the model. Despite that, we are including the results from the night-time inversion because the residuals for the nighttime analysis compared to the daytime analysis still appear to show good agreement, as seen in Fig. 9. Additionally, the nighttime is of particular interest to this investigation, as the primary source of $PM_{2.5}$ emissions in Christchurch is home heating (Sect. 3.2), which are typically used in the evening when the ambient temperature is at its lowest. Thus, the results from the nighttime inversion are presented here as well, with the note that they should be accepted with caution.

The nighttime posterior values inferred by the inversion and shown in Fig. 9 indicate generally more variable emissions from night-to-night compared to the results from the daytime inversions. Some of this variability can be explained in part to originate from the transport model uncertainty related to the nocturnal boundary layer height, as explained above. In general, however, the result present in the daytime inversions, i.e. that the posterior estimates are significantly lower than the corresponding priors, is upheld.

Figure 10 shows two time series plots from measurement sites towards the center of the domain (near the city center). By multiplying the prior or posterior flux maps by the transport operator **H**, we can compare the prior and posterior estimates against the measurements directly in concentration-space. Then, for any given measurement site, these time series can be plotted. By noting where the red line (posterior estimates) matches the measurements (black line) better than the blue line (prior estimates), we see the adjustments by the inversion at this measurement site.

In general, the concentrations derived from the posterior converge from the prior concentrations towards the measured enhancements, which indicates that the inversion is working insofar as the uncertainty on the measurements is much smaller than that on the prior fluxes, and therefore the measurements have a strong influence on the posterior result. The model has difficulty with capturing some of the strong spike events in the measurements, especially during nighttime, but overall it is able to follow the general variability seen in the enhancement time series, across all instruments (Fig. 10). The events of high

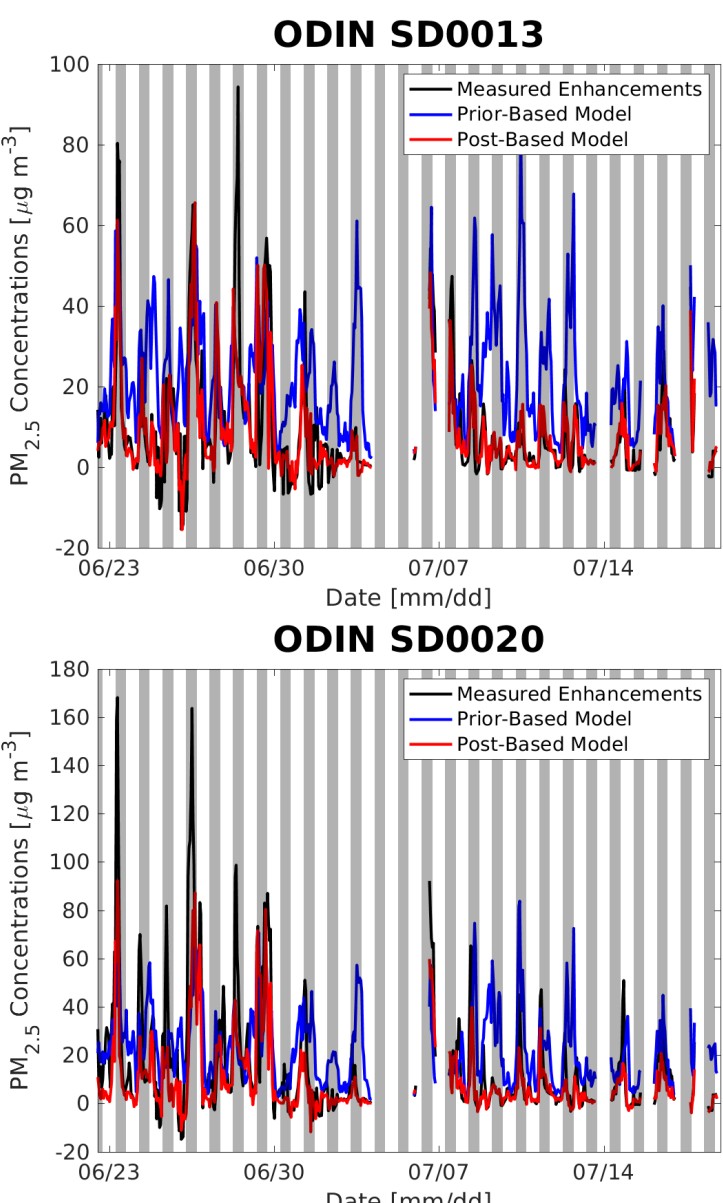

**Figure 10.** Example time series for in-city measurement sites showing the prior-based concentrations, the measured enhancements (i.e. PM$_{2.5}$ concentration measurements where the background has been removed), and the posterior-based concentrations, showcasing the improvement provided by the inversion. Grey shaded areas indicate nighttime hours.

545    PM$_{2.5}$ concentrations correspond to periods of low wind speeds and low temperatures. However, these enhancements could also be the result of a local source, which either is not well-resolved by the inverse model due to the comparatively-coarse 1-km resolution or is a consequence of having instruments positioned too close to the surface.





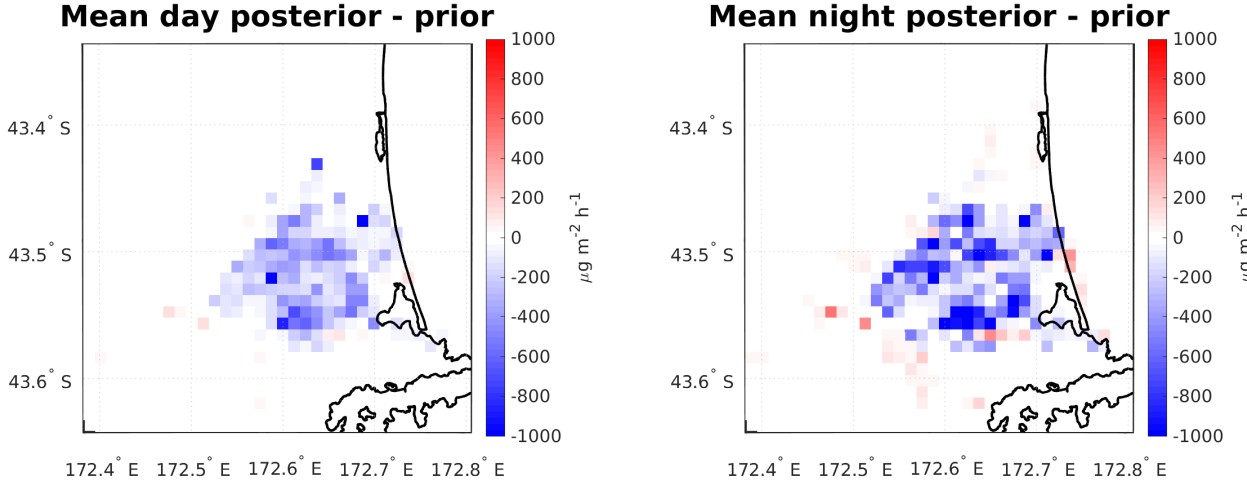

**Figure 11.** The mean difference between the posterior and prior maps in the daytime (left) and nighttime (right) across the full inversion period. The mean prior and posterior were calculated from the 10 daytime and nighttime maps, respectively.

To understand where the adjustments are being made spatially, we include maps of the mean difference between the posterior and the prior over the course of the analysis period, for both daytime and nighttime, as shown in Figure 11. The correction for apparent overestimates in the prior is distributed throughout the city during both day and night. Also, several of the large point sources present in the prior (see the 'true flux' map in Fig. 7), have had much of the corrections attributed to them. This is a consequence, in part, of the flux uncertainty scaling with the prior emissions, as the inversion will preferentially attribute corrections to grid-points with large uncertainties.

Furthermore, the results reveal an apparent increase in emissions in a ring just outside of the city, especially during nighttime. Given that these areas outside of the city had fewer measurement sites nearby, and thus fewer footprints overlapping the regions with which to issue corrections, it is possible that these sites are simply reflecting under-representation in the model. This may be evidenced by the fact that these effects are amplified at night, where the transport error is at its highest due to the difficulties with accurately simulating the nocturnal boundary layer. However, it is also possible that these are real sources of emissions that are not well-captured and identified by the inventory.

The results of the presented real-data study are considered to be providing an improved estimation of the quantity and distribution of $PM_{2.5}$ emissions from the region during the wintertime 2019 measurement campaign. However, the study still contains several substantial sources of uncertainty that may have affected the results and should therefore be acknowledged. Although they have been identified throughout this paper, we restate them here for clarity and transparency. First, the low sampling heights for the majority of the instruments recording in situ measurements (46 % of instruments at or below 3 m above the ground) open the door for local turbulence to have impacted the measurements; future studies should make every effort to record measurements at higher altitudes. Next, the modeled atmospheric transport always involves some potentially-impactful amount of uncertainty. Beyond the characterized mismatches with the winds, a future study could become more





robust by including in situ measurements of the boundary layer, as well. Additionally, the characterisation of the inflow, used to define the background PM$_{2.5}$ values across the measurement stations at any particular hour, can be a significant source of
uncertainty, as well.

## 6   Conclusions

The PM$_{2.5}$ inversion system set up through the MAPM project using Christchurch, New Zealand as its test-bed has been shown to be an effective system for assessing aerosol emissions in the urban environment using a large number of (i.e. 49) measurement sites. Eight OSSEs were performed to evaluate the usefulness of having a comparatively larger number of instruments
with a high uncertainty or a smaller number of instruments with a low uncertainty. Additional OSSEs were performed to test the effectiveness of using a time-evolving prior emission map, which uses the posterior flux map derived from the inversion on day $n$ as the prior emission map on day $n + 1$. Finally, the system was applied to real measurements taken during the MAPM field campaign during the winter of 2019.

The first eight OSSEs have shown that a lower number of instruments with low uncertainty will out-perform a higher number
of instruments with high uncertainty. While the addition of instruments with any uncertainty characteristics (large or small) does have some positive impact on the overall performance of the inversion, this impact is very small, especially in cases where the uncertainty is also very high. These results suggest that for future projects that may have to choose between these options, investing in a few instruments with small uncertainties might be more beneficial than having a larger number of sensors with high uncertainties. However, this may also depend on the city's design, and therefore these OSSEs should be repeated in any
other city where MAPM is going to be deployed, before sensors are set-up.

The results from the additional set of OSSEs, using the time-evolving prior, indicate that further tests need to be run before any firm conclusions can be drawn. Our initial tests have uncovered an apparent feedback mechanism that brings the posteriors farther away from the truth over time, yet it is still not clear what that mechanism is. If this process is able to be isolated and marginalized, then the methodology may be viable.

The inversion results from the real-data analysis suggest what appears to be a systematic overestimation in the prior emissions estimates of PM$_{2.5}$ for Christchurch, which is in the range of 40–60%, but which may be higher due to limitations of the inverse calculation in situations with such a large mismatch. This conclusion is in line with what would be expected, considering several of the assumptions that had been used in the calculation of the inventory on which the prior emissions estimate is based. The fact that the prior emissions estimate was created with an eye towards a worst-case scenario, in order to err on the side of
caution for human health, means that an overestimation of some significance would have been expected. Site comparisons against the measurements in concentration-space appear to give further support to the idea that the inversion is capturing the real trends seen in the measured enhancement data.

The presented results constitute a proof of concept study for inferring PM emissions sources on city scale, using Christchurch as a test-bed for high-measurement-density aerosol urban inversions, to provide city officials with near real-time assessments
of surface emissions. We successfully demonstrate the feasibility of the system and our investigation has itself identified



potential areas of improvement in the current emissions estimates for Christchurch. This study lays the groundwork for future investigations that may seek the same goal in disparate urban environments around the globe.

We assessed the different sources of uncertainty inherent in our approach. One of the largest areas is the transport model. Thus, the logical next step is going to be to implement a higher-resolution model that can better characterize wind flows and
turbulence at urban scales: the PArallelized Large-eddy simulation Model (PALM; Maronga et al., 2015). This will be coupled to WRF and, eventually, will include chemistry in the model to capture the effects of aerosol chemistry and deposition. The coupled WRF-PALM system, which has been developed by Lin et al. (2020), will be better able to simulate air flow (including, for example, effects of turbulence and diffusivity) in complex urban terrains, and would provide higher-resolution output (10 m or finer, compared to the current 1 km resolution).

*Code availability.* The Matlab code for the inverse model described in this paper is hosted on GitHub and access can be obtained from the corresponding author.

*Data availability.* The PM data collected during the MAPM field campaign are publicly available from https://doi.org/10.5281/zenodo.4023402 (Dale et al., 2020a). AWS data that were collected by the permanently installed AWSs are available from NIWA (https://cliflo.niwa.co.nz/) or the New Zealand MetService.

*Author contributions.* BN wrote much of the inverse model code, the code for all OSSEs, and wrote much of the paper together with input from all co-authors. SK assisted with writing of the paper, performed the FLEXPART-WRF simulations, and participated in the discussions around the inverse model set-up and results. SM-F managed the inverse model development and set-up, provided critical input throughout the project regarding its set-up, feedback on which analyses to perform and how to present them, and edits for the paper. GB participated in many of the discussions regarding the OSSEs that were performed in support of this paper, wrote some sub-sections of the paper, and copy-edited
other sections of the paper. LB generated the bottom-up emissions map based on census data and participated in the discussions around the inverse model results. LB and DL performed the WRF simulations that were used in this study, while ED performed the comparison of the WRF output to observations. ED provided all PM and AWS measurements used in this study. GO and ES performed the uncertainty analysis on the measurements.

*Competing interests.* The authors declare no competing interest.

*Acknowledgements.* We acknowledge the New Zealand Ministry of Business, Innovation and Employment (MBIE) for funding the MAPM project under contract BSCIF1802. We additionally acknowledge Guy Coulson and Ian Longley from NIWA in Auckland, New Zealand



for their time and efforts contributing to this project, particularly in treating the measurement data. We further acknowledge Tim Mallett from Environment Canterbury for his assistance with the creation of what became the prior emissions maps, as well as his insight and contextualisation of our results. Finally, we acknowledge Beata Bukosa from NIWA in Wellington, New Zealand, who attended the regular

(near-weekly) progress/update meetings and joined in the discussion or offered her perspective at several points throughout the process.





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
