# Peer review of "The MAPM (Mapping Air Pollution eMissions) method for inferring particulate matter emissions maps at city-scale from in situ concentration measurements: description and demonstration of capability"

_Atmospheric Chemistry and Physics, 2020_

## Referee Comment (RC2)

**General Comments**

This paper describes and demonstrates a system for estimating PM2.5 emissions given concentration measurements. This classical inverse approach has been widely used for greenhouse gases, less commonly used for chemically reactive gases and rarely for aerosols. The reasons for this are two-fold: Firstly the modelling of aerosol transport and modification/removal is difficult so introduces potentially confounding uncertainty into the calculations. Furthermore this uncertainty is hard to characterise, making the specification of the required PDFs difficult. Secondly the specification of background concentrations is difficult.

The authors have noted the background concentration problem though I think their method of dealing with it is debatable. They haven't dedicated as much attention to the aerosol modelling problem or its impact on their calculations.

The paper is clearly written with the methods fairly thoroughly described. It is certainly within scope for the journal.

Since this is a comment rather than final review I will concentrate on more general points.

**Major Points**

**Modelling Aerosol Transport and Production/Loss**  The study treats PM2.5 as an inert tracer with no explicit surface loss processes (more on this below). Is this true? It's clearly a question of scale. Probably the right question is whether the transit time is short compared to the atmospheric lifetime. At the least it should introduce an extra uncertainty into the problem. The authors remove observations strongly influenced by scavenging which removes one important process. Surface deposition, though remains. If it is significant it might help explain the findings of the "real data" experiment. The inversion solves for net surface fluxes, a combination of sources and deposition. The prior, using only sources, misses part of the story. Could part of the reduced posterior flux be the influence of deposition? Finally there is no treatment of secondary aerosol. This is regarded as a significant contribution elsewhere so its importance should be at least discussed.

**Background Concentration**  As the authors note, this is a bugbear for most regional inverse studies. The authors have chosen an elaboration of the "upwind background" approach often used. This requires that the aerosol field in the background air is homogeneous enough that the contribution of the background can be described by a single site (or a single site per wind direction here). It's pretty easy to construct cases where this won't hold, e.g. a source close enough to the city perimeter that its plume is missed by the background site but seen by the sites used in the inversion. The real world probably contains less artificial versions of this problem and we can't really tell how serious they are. There are approaches which treat the boundaries explicitly (e.g. Ziehn et al., 2014) but these depend on the model's ability to calculate sensitivity with respect to boundary cells. In any case the background should not be ignored in the OSSEs where errors in the background concentration will introduce correlated errors into the enhancements which should be treated in the observational covariance.

**Minor Points**

**Negative Fluxes**   the solution method used by the authors is the classical linear approach of Tarantola (2005). This is quite capable of producing negative fluxes. Does it do so here? If so how are they treated? One should not truncate the emissions as positive post hoc since the resultant emissions map no longer minimises the Bayesian cost function. One can solve the problem subject to a positive constraint but this usually applies iterative solutions rather than the one-shot matrix method described here.

**Calculation of Uncertainties**   The authors describe a Monte Carlo approach for doing this. If they are using a pure matrix method for the Maximum Likelihood Estimate (MLE) solution they don't need such an approach, the direct solution of the posterior covariance will work better. Furthermore it will not introduce potentially spurious correlations arising from the small ensemble size (usually ten in this study I think). the fact that the authors do use this Monte Carlo approach suggests there is some kind of positivity constraint for the MLE but more explanation is warranted.

**Regularisation**   On P13 the authors describe the use of exponential correlation functions for their prior emissions. The justification is that these are needed over and above the specification of a prior covariance i.e. the Bayesian approach itself) in order to regularise the problem. I think this is unnecessary and not well supported by the reasons given. The Bayesian problem in the linear Gaussian case should always return a solution. Its uncertainty might be large and allow for an MLE that doesn't look very nice. That's a pity but still a fair statement of what the data allows us to say. the prior covariance shouldn't be used as a numerical device to avoid this. Rather, as required by the Bayesian formulation, the prior covariance should encapsulate the prior PDF for emissions. Positive covariance between neighbouring grid cells implies that an error of the *prior* estimates in one grid cell suggests an error of the same sign in neighbouring grid cells. Just as likely is an error in grid cells governed by the same emissions process, no matter how far they are away. I would like to see at least one test case where the prior covariances were removed.

**References**

Tarantola, A.: Inverse problem theory and methods for model parameter estimation, no. 89 in Other Titles in Applied Mathematics, Society for Industrial and Applied Mathematics, 2005.

Ziehn, T., Nickless, A., Rayner, P. J., Law, R. M., Roff, G., and Fraser, P.: Greenhouse gas network design using backward Lagrangian particle dispersion modelling: Part 1: Methodology and Australian test case, Atmospheric Chemistry and Physics, 14, 9363–9378, doi:10.5194/acp-14-9363-2014, URL http://www.atmos-chem-phys.net/14/9363/2014/, 2014.

---

## Author Response (AR1)

**Response to reviews**

May 21, 2021

**1  Anonymous Referee #1**

This paper aims to both demonstrate the feasibility of an aerosol sensor network - inversion system and to use that system to quantify aerosol emissions in Christchurch during winter. The paper is interesting and relevant to the ACP readership. I recommend publication after attention to the issues outlined below.

Thank you very much for your review. We are pleased that you think our manuscript merits publication after addressing your concerns. We will do our best to adequately respond to each concern raised. The reviewer comments are repeated below in blue, with our detailed reply in black.

**Major comments**:

1. The description of construction of the background aerosol is difficult to follow. I recommend it be moved to the supplement so as not to disrupt the flow of the paper and expanded with additional figures that would clarify the approach.

   On this point, there is unfortunately a disagreement between the reviewers. The second reviewer has commented on the importance of including a rigorous discussion on this point. We, the authors, tend to agree with the second reviewer on this topic. However, we agree with this reviewer that the description about the construction of the background concentrations was rather confusing and therefore we have comprehensively rewritten the section for clarity.

2. The authors might be saying that the only important aerosol emissions are from wood burning and that this source is the entirety of their prior emission inventory. If this is the case, a clearer statement would help.

   Wood burning is the largest source of PM in Christchurch during winter, but emissions from traffic and industry are also included in our prior, as described on lines 234/235 of the original manuscript. Wood burning accounts for 72% of our prior emissions across the whole of the domain, while traffic and industry emissions contribute 20% and 8% respectively.

   An overview of the importance of wood smoke as a pollutant in New Zealand is given in Coulson et al. (2017) and Wilton (2014). Wood-smoke was identified as a problem in New Zealand as long ago as 1970 and a major study was carried out in the Hutt valley near Wellington in the late 1970s and early 1980s to investigate whether it posed a threat to health (Wratt et al. 1984). Several source apportionment studies indicated that PM from wood-burning is a major contributor to elevated winter-time concentrations of PM in various locations around New Zealand (Scott 2006; Davy 2007; Wilton et al 2007; Davy et al 2011, Tunno et al 2019). We now include a number of these references in the Section 2 of the revised manuscript.

3. Also, I found it odd that the prior is unrelated to temperature as I'd expect that heating would be needed more on cold than on warm days (relatively speaking).

   While the reviewer raises a good point that the emissions prior may be improved by the incorporation of a temperature dependency, we did not have sufficient Christchurch-specific bottom-up data in this investigation that would allow for such a construction. Further, in order to be performed well, such a task should require the

use of complex modeling, which is outside the scope of our current investigation. We have added a sentence to the manuscript to acknowledge that such an addition would be an improvement in the future.

**Material added to the paper (at the beginning of Section 3.2):**
At the time of this study, there was insufficient Christchurch-specific bottom-up data available to incorporate a temperature dependency into the prior, however we acknowledge that a robust handling of this could lead to improved posterior estimates, and should thus be considered in future investigations.

4. Since the inverse is a spatial map, it would help to provide a spatial map of the prior.

   A spatial map of the prior emissions is provided in the former Figure 5 of the manuscript.

5. An analysis of the temperature dependence in posterior emissions would help to indicate that the inverse has produced a reasonable result.

   This is a good idea. The results are shown below. In Figure 1, the temperature time series obtained from an AWS located near the city centre (Kyle Street) is plotted together with the sum of the inferred day-time (6 am to 6 pm NZST) and night-time (6pm to 6 am NZST) $PM_{2.5}$ emissions. The posterior emissions are, as expected, anti-correlated with temperature, i.e. during night-time when the temperature is low, the emissions increase and vice versa. Also, during warm periods, e.g. between the 2nd and 5th of July, while following a diurnal cycle, the emissions are rather low, reflecting that wood-burner usage was reduced during that period. These results give us confidence that the inverse model is producing reasonable results. This analysis and a similar paragraph have been added to the Results section of the manuscript.

[Figure]

[Figure]

Figure 1: Hourly mean temperature as obtained from the AWS located at Kyle Street, together with the sum of the inferred PM$_{2.5}$ emissions for day-time (blue) and night-time (green).

6. I suggest that the scope of this paper be narrowed to an explanation of the inversion system and analysis of inversion results. The work concerning OSSEs, while interesting, should either be placed in supplemental information or expanded upon and turned into a separate manuscript. As written it distracts from the paper and is not well developed.

After consultation with the co-authors, we agree with the reviewer, and the OSSE analyses have been removed in the revised manuscript. Following the suggestion by the reviewer, we will expand on our OSSEs and will publish the description and results in another publication, including a more comprehensive analysis. During the preparation of the additional publication, we will consider the comments provided by both reviewers here.

**Minor Comments**:

While I have many comments, the authors should adopt those they think will improve the paper without excessive additional effort.

1. Introduction - Introduction should be more succinct. Since Christchurch is not a megacity, discussion of megacities not irrelevant. Judgements about the correctness of epidemiological studies unnecessary.

   The discussion of mega-cities and the impact of air quality on health forms the motivation for this study and for why we developed a tool that can be used to infer PM emissions on a city scale, so that this method could be applicable to other cities. As a result, we would like to retain the material in the introduction. Christchurch was our target city used to develop the tool, and while not a mega-city it provides the environment that is used to describe the proof of concept of the methodology developed.

2. Introduction - The MAPM network is in many ways more similar to BEACO2N than INFLUX. Should cite Turner (2016 and 2020) in addition to Lauvaux.

   We have included the references as suggested by the reviewer.

3. Section 2 - More information (in a table) on instrument capability would be helpful. (Instrument precision, size distribution cutoffs at the high and low end, time resolution of data). Comment on whether it matters that the sensors you are using are missing the low end (less than 0.3 microns) of the aerosol size distribution and how much you think this impacts your source inversion.

   We agree with the reviewer that more information about the instruments is useful and therefore we have included additional material to the manuscript (see below). As the reviewer correctly points out, optical particle sensors have poor performance with aerosols smaller than 300nm, this being particularly true for the low-cost devices used here. Nevertheless, because the relevant metric in this study is $PM_{2.5}$ – i.e. the mass of particles smaller than 2.5 microns – it is worth noting that the mass size distribution of ambient aerosols, other than right next to high temperature combustion sources, is dominated by accumulation mode particles (size between 100nm and 1000nm). Therefore, it is expected that optical instruments perform well when measuring $PM_{2.5}$. In fact, the data from the co-location period – i.e. the period when the ODIN and ES-642 were compared against the standard compliant Tapered Element Oscillating Microbalance Filter Dynamics Measurement System (TEOM-FDMS), which is an established regulatory grade instrument (i.e. the TEOM meets the international standards for measuring PM), shows that the instruments compare well with a standard $PM_{2.5}$ monitor. So, even though the optical sensors do not 'see' the ultrafine particles such as those found within a few metres from high temperature combustion sources (e.g. traffic), the devices used in this study were not located in the immediate vicinity of traffic sources and they did show good agreement with standard $PM_{2.5}$ measurements. Therefore, we are confident that the impact of their size cut-off in our source inversion is minor.

   **Material added to the paper:** "The ES-642 has a stated particle size sensitivity of 0.1 to $100\mu$m with optimal response between 0.5 and $10\mu$m. The sensor has a prescribed accuracy of $\pm5$ % and a sensitivity of 1 $\mu$gm$^{-3}$ (Met One Instruments, Inc, 2019). The ODINs measure particles between 0.3 and $10\mu$m, with a counting efficiency of 98% for particles greater than $0.5\mu$m (Bulot et al, 2019). While the ES-642s made instantaneous observations approximately every second, which are then averaged to one minute resolution by the internal software, the ODINs took a single instantaneous measurement every minute."

4. Section 2 - Some recent studies have shown that a significant fraction ($\frac{1}{2}$) of urban aerosol enhancement can be secondary in nature. Discuss in more detail why these are not relevant (if they are not) to winter in Christchurch.

   We acknowledge that this is something that we should have discussed in the manuscript. Secondary aerosols are a small problem throughout New Zealand. New Zealand has very low background concentrations of precursors for secondary aerosol such as $SO_2$ (Coulson et al. 2016), and therefore, there is little opportunity for aerosol formation. Generally, secondary particulate is higher in summer because conditions are more conducive to

photo-chemistry (warmer, sunnier, etc.) especially for sulfates. The latest studies around secondary aerosol and their contribution to the total measured $PM_{2.5}$ is based on a 2014 report by Environment Canterbury (ECan) (`https://api.ecan.govt.nz/TrimPublicAPI/documents/download/2119173`). The report states that based on a study performed in Christchurch in 2001/2 they found that, on average, secondary aerosols were the source of only 14% of measured wintertime particulates, and that they had contributed only 5% to the measured wintertime maximum. These numbers are well below the 50% value that had concerned the reviewer, and should fall comfortably within the uncertainty bounds ascribed in our analysis.

We have added material to the revised paper that refers to secondary aerosol, explains why they have not been included, and acknowledges that this is a potential source of bias in the study.

5. Section 3.1 - Figure 2 schematic is disconnected and confusing. Modify to show connections between FLEX/PART - WRF, Prior, and Enhancements and how these are fed into the system to generate posterior

Thank you for catching that. The figure was not complete as some arrows went missing when converting the paper into pdf format. We corrected that and included an update of Figure 2, now showing all arrows and connections between FLEXPART model output and required inverse model inputs.

6. Section 3.1 - Provide a brief description of WRF settings, model-measurement meteorological mismatch, etc in the main text and move the bulk of this to a supplement.

We agree with the reviewer that the information provided about the WRF model might disturb the flow and is not essential to understand the main message and outcomes of this paper. We have therefore followed the suggestion by the reviewer and moved the majority of the material of section 3.1 into an appendix. We prefer to move the material into the appendix, rather than into a supplement. While the information is not critical to support the conclusion of the paper, it provides extra detail and/or support useful for experts in the field and therefore, according to the ACP guidelines, this material is better suited for an appendix, rather than supplement.

7. Section 3.1 - 185-214 does not belong in the methods section. Condense and move to introduction, or delete and use as part of a separate paper that includes OSSE experiments.

We agree with the reviewer that the material is be better suited for the introduction. We have shortened the section and moved the material into the introduction.

8. Section 3.1 - It would be nice to see overlay of footprints with sensors in Fig 1 or elsewhere.

We include here a $log_{10}$-scaled figure of the cumulative footprints for the Daytime and Nighttime inversions, respectively, in Figure 2. We decided not to include this additional information in the paper, because we do not think that it significantly adds to the analyses, and because the response to the referee comments accompany the paper such that readers who are interested in a deeper validation may refer to our response.

9. Section 3.1 - Put Figure 3 in a supplement. Important information, but not critical to your story arc.

We followed the suggestion by the reviewer and removed Fig.3 from the main paper. As mentioned above, we have now included an appendix and we have moved the Figure to the appendix.

10. Section 3.1 - Put Figure 4 in a supplement. Important information, but not critical to your story arc.

We followed the suggestion by the reviewer and removed Fig. 4 from the main paper. As mentioned above, we have now included an appendix and we have moved the Figure to the appendix.

11. Section 3.2 - 235 - Provide sources for percentages in this paragraph.

Environment Canterbury (ECan, Tim Mallett) provided the percentages cited here which were used to create the prior emissions map. ECan develops PM emissions inventories including emissions from home heating, industries, and traffic:

[Figure]

[Figure]

Figure 2: Log$_{10}$-scaled figures showing the total cumulative footprints from all of the influence functions during the daytime (left) and nighttime (right) inversion periods. The tower sites are marked with red circles.

(a) Emissions from home heating are estimated by applying emission factors to home heating activity data which are typically obtained from census data. Data collected include number, type, and age of appliance and fuel use for all appliances. Home heating emission factors are defined as the mass of pollutant discharged per unit mass of source material and are usually expressed as grams per kilogram (e.g. ECan 2014 technical report, (`https://api.ecan.govt.nz/TrimPublicAPI/documents/download/2119173`)). Emissions factors are updated regularly and new census data are used, when they become available, to generate regular updates of emissions inventories.

(b) Emissions from traffic were estimated using activity data in the form of vehicle kilometres travelled for various congestion conditions or levels of service, and the application of emission factors to these data. Emissions factors are calculated using the Ministry of Transport New Zealand Vehicle Emissions Prediction Model (`https://www.nzta.govt.nz/roads-and-rail/highways-information-portal/technical-disciplines/air-quality-climate/planning-and-assessment/vehicle-emissions-prediction-model/`).

(c) Emissions from industries were estimated using survey data that were collected by ECan. Industries that generate air discharges were identified using a combination of ECan resource consent files and the use of telephone directories and each identified business was mailed a survey. The industrial emission factors used were primarily derived from the USEPA AP-42 compilation of emission factors, with additional data from Air Chief 12 (USEPA 2005).

The emissions factors are updated regularly and ECan is about to publish the 2018 inventory on which the numbers cited in this paper are based. However, at this time, the inventory has not been published and therefore we can not provide a reference other than personal communication with Tim Mallett at ECan.

12. Section 3.2 - 248 - Where does this 75% come from? Why is it reasonable?

First we note that the stated 75% was an error. It should have been 73% and this has now been corrected in the manuscript. 73% is an estimate based on the information available, i.e. a comparison of census data, the number of households that use wood/coal to heat their house, and the chimney survey, providing information about what percentage of all chimneys were actually hot on a given night. As such, the 73% estimate is an expert-judgement derived value by ECan based on the information available.

13. Section 3.3 - Show time series of background against other measurements to persuade the reader that background method makes sense.

We include in Figure 3 a time series comparison of the adjusted background value against two of the measurement sites located near the center of the city. These have been chosen to be sites SD0013 and SD0020, to be consistent with the former Figure 10. Here we are able to show that the new presented methodology creates a background time series that does not appear to contain any of the large spikes from stochastic local events, while

still containing real, apparently-reasonable background variations. This figure and a slight rephrasing of this paragraph have been added to the manuscript in the reworked Background portion of the Methodology section.

[Figure]

Figure 3: A time series comparing the adjusted background value against two measurement sites located near the city center.

14. Section 3.3 - The final step, describing the 10 day median is confusing and could use justification. If PM enhancements are synoptic in nature, this step does not make sense.

We are unclear about the second sentence, but we attempt a response anyway. The 10-day median smoothing came out of several different filtering and averaging attempts we had made to try to minimize apparent stochastic local influences from our background signal. (A deeper explanation of the process of the rest of the formulation of the background time series can be found in a response to the second reviewer.) Since the duration of synoptic events can vary from the scale of hours to weeks, we wanted to choose a period of time that would be able to encompass and represent effects from all circumstances. While at the stage where we were deciding on the most appropriate final smoothing to implement, we had additionally tested rolling averages and several low-pass filters, as well, before ultimately settling on the 10-day rolling median as our preferred method. Although, by this point of the process, we had found that there was not a large difference between the ultimate background characteristics resulting from many of these different smoothing approaches. We acknowledge that a different scientific team may have found a different technique to be preferable, but we do not expect that the results of the inversion analyses would be strongly affected.

15. Section 3.3 - Put wind direction thresholds in a table, rather than directly in text.

We followed the suggestion by the reviewer and included the wind direction angles in a table.

16. Section 3.4 - Add a sentence justifying choice of length scale.

The explanation of the choice of the length scale has been expanded, also in response to comments by the second reviewer.

17. Section 4 - Remove this section from this paper and put in supplement or expand and write a second paper.

We have decided to remove the OSSE analyses from this paper and to include a more comprehensive investigation in a future publication. The comments provided with respect to this section will be taken into account during that process.

18. Section 4.1 - Since error is introduced into the real inversion via choice of background value, it seems you are under-representing error by excluding background from OSSE. I suggest finding a way to compensate for this.

As stated, this section has been removed following the reviewer's recommendation.

19. Section 4.1 - Explain error introduced into the "true" time series more clearly. Does 10%/10% mean that for a "true" reading of 25 micrograms/m$^3$, we are sampling from a normal distribution of sigma 2.5 micrograms/m$^3$? Does 90%/90% mean that for the same "true" value, we are sampling from a normal distribution with mean 25 and sigma 22.5 micrograms/m$^3$? I am unclear from the text.

As stated, this section has been removed following the reviewer's recommendation.

20. Section 4.1 - The tradeoff between a large number of high-precision and a small number of low-precision instruments in this type of system is important. However, if my interpretation of 90%/90% is correct, the experiment set up here contrasts relatively good sensors with unrealistically poor sensors. As a result, I do not believe your set-up answers the question in a realistic way. The error you choose in this experiment should be reflective of real sensors (ideally those used in the MAPM project). If you believe the error you have chosen is actually reflective of these sensors, please document this in the text. See also Turner et al 2016.

As stated, this section has been removed following the reviewer's recommendation.

21. Section 4.2 - This is an interesting idea that is worth pursuing, but the results presented here are not adequate for publication. I wonder whether adapting prior based on a longer time average of posteriors would be a better idea.

As stated, this section has been removed following the reviewer's recommendation.

22. Section 5 - Figure 9 - Based on Lauvaux (2020), we expect time series (individual hours/ days) of posterior to show random error, but posterior averaged over a longer time period to be more meaningful. We see this in your timeseries, making it difficult to interpret. Perhaps add a median diel cycle of posterior with error bars to make it easier to see how posterior compares to prior on average.

While we appreciate the reviewer's suggestion, we think that the fact that our analyses have been split up into separate "daytime" and "nighttime" inversions means that the inclusion of a median diurnal trend line would not provide much value to the reader. This would be exacerbated by the fact that there is only one "daytime" and "nighttime" posterior value for any given day. We would need to have a higher-time-resolution analysis in order to be able to construct a median diurnal posterior time series that would be sufficiently meaningful.

23. Section 5 - 506 - Earlier, you said 75%? Now 70%. Are these different numbers? Not clear from the text.

We note that we believe the reviewer is referring to the sentence starting '80% of all wood burners...' in the paper. The reviewer is indeed correct that this was a mistake. It should read 73% and we corrected that mistake in the revised manuscript. Note that the 75% was a mistake beforehand (as mentioned above). We also re-worded the sentence for clarification.

24. Section 5 - Comment briefly on whether you think the whole difference between prior and posterior is due to home heating or whether other sources might play a role. Comment on whether you see temperature dependence in the posterior.

We have added a sentence to this section that acknowledges that most, but not all, of the corrections are expected to be attributable to home heating.

25. Section 5 - 506-512 - References needed.

We acknowledge that the reviewer would like to see published references for the numbers provided and we note that we discuss the assumptions underlying the generation of the prior that are based on the knowledge ECan has obtained over years of monitoring. We do not have any references that we can provide here other than Tim Mallett from ECan, personal communication, October 2020. ECan is a local government environmental agency that provides a range of data on local air quality and emissions sources. The data collected by ECan are not always published in the peer-reviewed literature. These data are not collected for scientific studies but to

meet regulatory requirements. We are confident in the numbers and information provided by ECan as they are based on research and investigations into air quality monitoring.

26. Section 5 - Figure 11 - Overlay of footprint would be helpful in thinking about the exterior ring.

We feel that an overlay of the footprints on the former Figure 11 would decrease its readability more than it would help to inform the reader. We have included the cumulative footprint figures for the daytime and nighttime periods in a previous response (Figure 2). Since there is no apparent correlation there between footprint coverage and the anomalous signals around the outskirts of the domain of the city, we do not feel that the inclusion of this figure will materially enhance the manuscript. For now, we are choosing to leave it out of the manuscript, though it remains here in our responses, which are also accessible to a sufficiently curious reader.

27. Section 5 - 555 - Comment: What is actually in the space that shows higher emissions in posterior than in prior? Is it actually possible that there are unaccounted emissions of the magnitude shown: Are their homes or is it empty space? What are the values in that area in the prior?

While the prior, based on information from 2018, expects very low emissions in these areas, it is unknown whether there may have been new developments in the area after the creation of the prior. We have now expanded a sentence acknowledging this in addition to our characterization that it this area may simply not be as well constrained by footprints as the rest of the domain.

28. Section 5 - 564 - You comment on the possibility that the MAPM network is picking up hyper-local sources due to height. Use the MAPM time-series data to make an argument one way or another. Do the sensors detect signals propagating through the network? Do enhancements over background at various sites correlate or are they totally unrelated?

The idea that there is additional uncertainty to be considered in our analyses due to the measurement heights of our instruments is one that is commonly understood in the field of inverse modeling, because measurements should ideally be taken after emissions have become well-mixed. For example, Miles et al. (2017) found that differences in sampling height among the tower-mounted sensors (for them, ranging between 39 and 136 meters above ground level) in the INFLUX project (which monitored $CO_2$ over Indianapolis, IN) could affect the reported urban enhancement by as much as 50%.

In our case, since the emission sources would be expected to be a mixture of near-ground (e.g. traffic) and elevated (e.g. stack from a wood burner), the height and proximity of the sensors could have a noticeable impact on the measurements. While the reviewer's suggestion that we try to identify individual signals propagating through the data poses an interesting scientific question, a robust analysis of this sort would be challenging given the imperfect knowledge of the distribution of the potential sources at such local scales. We respectfully think that such an analysis would be both outside the scope of this manuscript and would fall into the category of what the reviewer described as unnecessary "excessive additional effort" at the beginning of this Minor Comments section.

29. Section 5 - Section 5 - Figure 10 - Time series is too squished. Expand figure horizontally or show us scatter instead.

These time series figures have been stretched out along their x-axis, to improve clarity. They are included here as Figure 4, as well, for reference.

30. Section 6 - Omit discussion of OSSEs.

The discussion of the OSSEs has been removed from this section.

**References**

A.J. Turner, J. Kim, H. Fitzmaurice, C. Newman, K. Worthington, K. Chan, P.J. Wooldridge, P. Köhler, C. Frankenberg and R.C. Cohen, Observed impacts of COVID-19 on urban CO2 emissions, Geophys. Res. Lett., https://doi.org/10.1029/2020GL090037, 2020.

[Figure]

Figure 4: Example time series for in-city measurement sites showing the prior-based concentrations, the measured enhancements (i.e. PM$_{2.5}$ concentration measurements where the background has been removed), and the posterior-based concentrations, showcasing the improvement provided by the inversion. Grey shaded areas indicate nighttime hours.

A.J. Turner, A.A. Shusterman, B.C. McDonald, V. Teige, R.A. Harley and R.C. Cohen, Network design for quantifying urban CO2 emissions: Assessing tradeoffs between precision and network density Atmos. Chem. Phys, 16, 13465-13475, doi:10.5194/acp-16-13465-2016, 2016.

**Additional references added to revised manuscript**

Bulot, F.M.J., Johnston, S.J., Basford, P.J. et al. Long-term field comparison of multiple low-cost particulate matter sensors in an outdoor urban environment. Sci Rep 9, 7497 (2019). https://doi.org/10.1038/s41598-019-43716-3

Coulson, G., Olivares, G., and Talbot, N. (2016) Aerosol Size Distributions in Auckland. Air Quality and Climate Change Volume 50 No.1. February 2016

Coulson, G., Somervell, E., Mitchell, E., Longley, I., and Olivares, G., (2017) Ten years of woodburner research in New Zealand: A review. Air Quality and Climate Change. Volume 51, No. 3, December 2017.

Davy, P., K., Trompetter, W. J. and Markwitz, A.: Source apportionment of airborne particles in the Auckland region, GNS Science Client Report 2007/314, 2007.

Davy, P., K., Trompetter, W. J. and Markwitz, A.: Source apportionment of airborne particles in the Auckland region: 2010 Analysis' GNS Science Consultancy Report 2010/262, November 2011.

Jorn D. Herner, Qi Ying, Jeremy Aw, Oliver Gao, Daniel P. Y. Chang & Michael J. Kleeman (2006) Dominant Mechanisms that Shape the Airborne Particle Size and Composition Distribution in Central California, Aerosol Science and Technology, 40:10, 827-844, DOI: 10.1080/02786820600728668.

Scott, A.: Real life emissions from residential wood burning appliances in New Zealand, Ministry for the Environment Sustainable Management Fund report, ID number 2205, 2005.

Tunno, B., Longley, I., Somervell, E., Edwards, S., Olivares, G., Gray, S., Cambal, L., Chubb, L., Roper, C., Coulson, G., Clougherty, J.: Separating spatial patterns in pollution attributable to woodsmoke and other sources, during daytime and nighttime hours, in Christchurch, New Zealand, Environmental Research, 171, Pages 228-238 doi: https://doi.org/10.1016/j.envres.2019.01.033, 2019.

Wilton, E., Smith, J. and Davy, P., Source identification and apportionment of PM10 and PM2.5 in Hastings and Auckland, NIWA report CHC2007-137, 2007.

Wilton, E., 2014 Review - particulate emissions from wood burners in New Zealand. Emily Wilton. `https://www.niwa.co.nz/sites/niwa.co.nz/files/WoodburnerReportFinal.pdf`

Wratt, D.S., Salinger, M.J., Clarkson, T.S., Imrie, B.W., Bromley, A.M., Lechner, I.S., 1984 Airflow and dispersion in a valley: The Upper Hutt Study. Scientific Report 4. NZ Meteorological Service. `http://docs.niwa.co.nz/library/public/NZMSSR4.pdf`

**2  Reply to Peter Rayner**

**General Comments**:

This paper describes and demonstrates a system for estimating PM2.5 emissions given concentration measurements. This classical inverse approach has been widely used for greenhouse gases, less commonly used for chemically reactive gases and rarely for aerosols. The reasons for this are two-fold: Firstly the modelling of aerosol transport and modification/removal is difficult so introduces potentially confounding uncertainty into the calculations. Furthermore this uncertainty is hard to characterise, making the specification of the required PDFs difficult. Secondly the specification of background concentrations is difficult. The authors have noted the background concentration problem though I think their method of dealing with it is debatable. They haven't dedicated as much attention to the aerosol modelling problem or its impact on their calculations. The paper is clearly written with the methods fairly thoroughly described. It is certainly within scope for the journal. Since this is a comment rather than final review I will concentrate on more general points.

We thank you very much for the review and look forward to addressing your concerns in the following paragraphs.

**Major Points**

**Modelling Aerosol Transport and Production/Loss**

The study treats PM2.5 as an inert tracer with no explicit surface loss processes (more on this below). Is this true?

In designing the inverse model we considered the following loss processes between the source and the receptor:

- Chemical transformation: the air chemistry in Christchurch in winter is such that there is little, if any, chemical transformation of PM, as New Zealand has very low background concentrations of $SO_2$ (Coulson et al., 2016). We therefore discounted this as a mechanism for loss of PM between the source and receptor.

- Wet deposition: to avoid the complications of wet deposition we only apply the inverse model during periods more than 12 hours outside of rainy periods.

- Dry deposition: we considered both gravitational deposition from particles growing in size and surface deposition from interactions with buildings, the ground, trees etc.. Because of the absence of chemical processing of the PM, there is little likelihood of the particles growing in size through chemical processes and gravitationally precipitating out of the air column. Loss through other dry deposition processes is possible but it is unlikely to

play a large role, since:

i) a study in California (Herner 2007) under similar wintertime inversion conditions modelled deposition rates for a 0.1 $\mu$m, a 0.5 $\mu$m, and a 3 $\mu$m particle to be $2.4 \times 10^{-4}$ ms$^{-1}$, $6.4 \times 10^{-4}$ ms$^{-1}$, and $4.34 \times 10^{-4}$ ms$^{-1}$ respectively. During the evening hours when mixing depths are on the order of 50 m, these velocities suggest deposition time scales of 58 h, 217 h, and 32 h for ultrafine, accumulation mode, and coarse airborne particles, respectively.

ii) Jeanjean et al. (2016) investigated the PM$_{2.5}$ reduction via deposition at a city scale (Leicester) and their results suggest that PM$_{2.5}$ deposition on buildings is negligible. Furthermore, deposition on trees and grassland is more important, yet a lot smaller in winter compared to summer, because the deposition depends on the available leaf area.

iii) Given the relatively small domain of Christchurch city, and our subsequent prescription of a 12-hour limit in transit time between source and receptor, we have assumed that there is insufficient time for much dry deposition to occur along the trajectory. The majority of the particles released in Flexpart leave the domain within the first few hours, with around 70% of the particles leaving within the first 5 hours.

iv) Most of PM$_{2.5}$ being measured in Christchurch is also smaller than $1\mu$m (see Figure 5). Very fine (sub 1 micron) particles have very slow deposition speeds, and hence long persistence in the atmosphere, and would therefore easily stay airborne while they were within a reasonable distance of the emissions (tens of kilometres).

ODIN_SD0172

[Figure]

Figure 5: A comparison showing the fraction of measured PM$_{2.5}$ that was PM$_1$ at one of the in-city measurement sites (41.5168° S 172.6150° E). *top left:* Scatter plot showing hourly mean PM$_{2.5}$ and PM$_1$ observations. *bottom left:* Angular histogram of hourly mean wind from direction measured at a nearby AWS (Kyle Street AWS; 43.5307° S 172.6077°). *top right:* histogram of hourly mean PM$_1$ as a fraction of PM$_{2.5}$. The colours indicate PM measurements made when the wind direction measured at the nearby Kyle Street AWS was in certain quadrants indicated in the figure legend. *bottom right:* The individual histograms that make up the histogram in the top right panel using the same colour scheme.

We recognise that these are untested assumptions for the city of Christchurch and (to our knowledge) no study has determined the dry deposition rates of PM$_{2.5}$ either from model simulations or from observations, so further validation is required in the future. A study by Saylor et al. (2019) has shown that deposition rates in air quality models are very uncertain and differ substantially between models, so it is clearly a subject for future research. For now, we acknowledge that ignoring this loss process could lead to a small under-estimate of the posterior emissions. If MAPM is applied in any other city, these loss processes will need to be reconsidered and, if necessary, appropriately included in the model.

It's clearly a question of scale. Probably the right question is whether the transit time is short compared to the

atmospheric lifetime. At the least it should introduce an extra uncertainty into the problem. The authors remove observations strongly influenced by scavenging which removes one important process. Surface deposition, though remains. If it is significant it might help explain the findings of the "real data" experiment. The inversion solves for net surface fluxes, a combination of sources and deposition. The prior, using only sources, misses part of the story. Could part of the reduced posterior flux be the influence of deposition? Finally there is no treatment of secondary aerosol. This is regarded as a significant contribution elsewhere so its importance should be at least discussed.

Yes, part of the reduced posterior flux could come from these processes that we have not included in the presented analyses, and we acknowledge this as a potential bias in the posterior emissions maps. A sentence has been added to the paper to this effect.

**Background Concentration**

As the authors note, this is a bugbear for most regional inverse studies. The authors have chosen an elaboration of the "upwind background" approach often used. This requires that the aerosol field in the background air is homogeneous enough that the contribution of the background can be described by a single site (or a single site per wind direction here). It's pretty easy to construct cases where this won't hold, e.g. a source close enough to the city perimeter that its plume is missed by the background site but seen by the sites used in the inversion. The real world probably contains less artificial versions of this problem and we can't really tell how serious they are. There are approaches which treat the boundaries explicitly (e.g. Ziehn et al., 2014) but these depend on the model's ability to calculate sensitivity with respect to boundary cells. In any case the background should not be ignored in the OSSEs where errors in the background concentration will introduce correlated errors into the enhancements which should be treated in the observational covariance.

Thank you very much for the comment. Indeed, we wanted to make clear the significant difficulty presented by finding a proper definition of the background. We initially started with a single background site assigned based on wind direction, but found that this was often not sufficient to eliminate apparent local influences. Next, we switched to a system wherein each wind direction had a group of potential upwind towers to choose from, where that choice was dependent on the standard deviation of the measurements during that hour, following the assumption that less deviation would imply less local contamination. Eventually, we even moved beyond this to the method presented, which tries to gather as much information as possible about the wind outside of the city being measured, to create a sort of ensemble-average per hour, which is then further smoothed with the 10-day rolling median to attempt to attenuate local influences.

The suggestion that the complexity of accurately capturing the background concentration, and the corresponding uncertainty it introduces into the problem, should be included in the OSSE simulations is a good one. Following additionally the comments and suggestions from the first reviewer, we have decided to remove the OSSE analyses from this paper, so that they may be presented in a more comprehensive manuscript submitted at a later date. During the preparation of that manuscript, these comments and suggestions will be considered.

**Minor Points**

**Negative Fluxes**

the solution method used by the authors is the classical linear approach of Tarantola (2005). This is quite capable of producing negative fluxes. Does it do so here? If so how are they treated? One should not truncate the emissions as positive post hoc since the resultant emissions map no longer minimises the Bayesian cost function. One can solve the problem subject to a positive constraint but this usually applies iterative solutions rather than the one-shot matrix method described here.

Yes, indeed the solution does provide several negative pixels in the posterior solution. We have imposed no positivity constraint, as we agree that that would be an inappropriate manipulation. Across the "daytime" inversions, a mean posterior map across each of the days yields 28 negative pixels out of the 9100 in the domain. Similarly, a mean of each of the nighttime posterior maps contains 29 negative pixels out of the 9100 in the domain. We have included a paragraph explicitly stating this in the manuscript now, so as not to unintentionally mislead the reader.

**Calculation of Uncertainties**

The authors describe a Monte Carlo approach for doing this. If they are using a pure matrix method for the Maximum Likelihood Estimate (MLE) solution they don't need such an approach, the direct solution of the posterior covariance will work better. Furthermore it will not introduce potentially spurious correlations arising from the small ensemble size (usually ten in this study I think). The fact that the authors do use this Monte Carlo approach suggests there is some kind of positivity constraint for the MLE but more explanation is warranted.

We believe that this comment is referring to the methodology of the presented OSSE analyses that attempt to chain together several days of inversions in a time-evolving fashion, rather than running them independently. As stated in a response to a previous comment, we have made the decision to remove the OSSE analyses from this paper, so that they can be included in a more comprehensive OSSE paper submitted at a later time. Comments such as these are greatly appreciated and will be considered during the preparation of that manuscript. The real-data inversions do not use any Monte Carlo approach, so we expect that the concerns raised in this comment should be satisfied.

**Regularisation**

On P13 the authors describe the use of exponential correlation functions for their prior emissions. The justification is that these are needed over and above the specification of a prior covariance i.e. the Bayesian approach itself) in order to regularise the problem. I think this is unnecessary and not well supported by the reasons given. The Bayesian problem in the linear Gaussian case should always return a solution. Its uncertainty might be large and allow for an MLE that doesn't look very nice. That's a pity but still a fair statement of what the data allows us to say. the prior covariance shouldn't be used as a numerical device to avoid this. Rather, as required by the Bayesian formulation, the prior covariance should encapsulate the prior PDF for emissions. Positive covariance between neighbouring grid cells implies that an error of the prior estimates in one grid cell suggests an error of the same sign in neighbouring grid cells. Just as likely is an error in grid cells governed by the same emissions process, no matter how far they are away. I would like to see at least one test case where the prior covariances were removed.

The reviewer raises a good point that the introduction of correlations should not be used simply in order to fix a mathematical complication. In our study, the spatial correlations are introduced in order to address the co-localization problem, wherein the grid cells that are co-located with measurement sites are given an unrealistically strong influence. Bocquet (2005) found that this problem gets worse as grid resolution increases, and defined the problem mathematically. Saide et al. (2010) further defined this issue and tested potential solutions to it, including the addition of spatial correlations, which are shown to at least help to address the problem.

Back to the reviewer's point, though, it would not be appropriate to include these additional correlations if there was no physical justification. There would, in fact, be a danger of creating new problems if the structures introduced were themselves unrealistic. To that end, we comply with the reviewer's request to showcase how these results change with and without the spatial correlations introduced. Figure 6 shows how the daytime and nighttime mean posterior/prior difference maps change if the spatial correlations are removed (left column) compared to when they remain (right column). We find that, at least by this metric, there is very little change in the difference maps. This is presumably because the solution is rather overconstrained by the large number of overlapping footprints from the receptor sites (see Figure 2 in a previous reviewer response). In this sense, it is a relief to see that the inclusion of the spatial correlations does not have a drastic impact on the solution, and that our conclusions would remain the same in either case.

This alone may not be enough, however, to justify that the inclusion is merited for real, physical reasons. Indeed, part of the justification for the inclusion of these spatial correlations parallels the reviewer's thoughts about how the error in grid cells governed by the same emissions processes should act similarly. Because most of the emissions are coming from wood-burners in residential homes, and because the residential areas tend to cluster near each other, the introduction of spatial correlations acts as a first-order attachment between these. We believe that the inverse relationship between the total posterior emissions and temperature, presented in Figure 1 in a response to the first reviewer, is evidence towards this end. As this second reviewer notes, such an inclusion will be insufficient to capture cases of sufficiently-spatially-isolated residential homes, for example. However we would not want to extend the correlation length too far, lest the entire domain be overconstrained in its degrees of freedom such that it could only correct emissions as a sort of scaling factor to a unified block. Thus, we believe that the addition of 1-km spatial correlations, with the error correlation length matching the size of the grid cells (although insofar as it is an exponential decay function, its influence will extend beyond), is appropriate. With that being the case, we acknowledge that we may be incorrect, and the correlations may be more complicated than we assume. So, we have updated the prior/posterior time series figure (formerly Figure 6) in the manuscript to also include as a dashed line the posterior solution without spatial correlations, in order to acknowledge that we may have overestimated the correction. Some

[Figure]

Figure 6: Posterior - Prior mean flux differences for daytime (top row) and nighttime (bottom row), comparing cases without the use of spatial correlations (left column) against those using a 1-km spatial correlation length (right column)

text has been added to explain this as well. The updated daytime and nighttime time series plots are included here, too, for reference, as Figure 7

Note: The authors consulted with Dr. Thomas Lauvaux for his expertise on this problem during the preparation of this response.

[Figure]

Figure 7: Updated time series of sum of the fluxes across the domain in the prior vs. posterior for the daytime and nighttime inversions. A posterior solution is also included for the case without the inclusion of spatial correlations, for comparison.

**References**

Tarantola, A.: Inverse problem theory and methods for model parameter estimation,no. 89 in Other Titles in Applied Mathematics, Society for Industrial and AppliedMathematics, 2005.

Ziehn, T., Nickless, A., Rayner, P. J., Law, R. M., Roff, G., and Fraser, P.:Greenhouse gas network design using backward Lagrangian particle disper-sion modelling: Part 1: Methodology and Australian test case, Atmospheric-Chemistry and Physics, 14, 9363–9378, doi:10.5194/acp-14-9363-2014, http://www.atmos-chem-phys.net/14/9363/2014/, 2014.

Bocquet, M.: Grid resolution dependence in the reconstruction of an atmospheric tracer source, Nonlin. Processes Geophys., 12, 219–233, https://doi.org/10.5194/npg-12-219-2005, 2005.

Jeanjean, A.P.R., Monks, P.S., Leigh, R.J.: Modelling the effectiveness of urban trees and grass on PM2.5 reduction via dispersion and deposition at a city scale, Atmospheric Environment, 147, 1-10, https://doi.org/10.1016/j.atmosenv.2016.09.033

Saide, P, Bocquet, M., Osses, A., and Gallardo, L.: Constraining surface emissions of air pollutants using inverse modelling: method intercomparison and a new two-step two-scale regularization approach, Tellus B: Chemical and Physical Meteorology, 63:3, 360-370, DOI: 10.1111/j.1600-0889.2011.00529.x, 2011

Saylor, R., Baker, B., Lee, P., Tong, D., Pan, L., and Hicks, B.: The particle dry deposition component of total deposition from air quality models: right, wrong or uncertain?, Tellus B: Chemical and Physical Meteorology, 71:1, DOI: 10.1080/16000889.2018.1550324